# Keypoint-Guided Optimal Transport with Applications in Heterogeneous Domain Adaptation

**Xiang Gu[1], Yucheng Yang[1], Wei Zeng[1], Jian Sun (✉)[123], Zongben Xu[123]**

[1] School of Mathematics and Statistics, Xi'an Jiaotong University, Xi'an, China
[2] Pazhou Laboratory (Huangpu), Guangzhou, China
[3] Peng Cheng Laboratory, Shenzhen, China
{xianggu,ycyang}@stu.xjtu.edu.cn {wz,jiansun,zbxu}@xjtu.edu.cn

## Abstract

Existing Optimal Transport (OT) methods mainly derive the optimal transport plan/matching under the criterion of transport cost/distance minimization, which may cause incorrect matching in some cases. In many applications, annotating a few matched keypoints across domains is reasonable or even effortless in annotation burden. It is valuable to investigate how to leverage the annotated keypoints to guide the correct matching in OT. In this paper, we propose a novel KeyPoint-Guided model by ReLation preservation (KPG-RL) that searches for the matching guided by the keypoints in OT. To impose the keypoints in OT, first, we propose a mask-based constraint of the transport plan that preserves the matching of keypoint pairs. Second, we propose to preserve the relation of each data point to the keypoints to guide the matching. The proposed KPG-RL model can be solved by the Sinkhorn's algorithm and is applicable even when distributions are supported in different spaces. We further utilize the relation preservation constraint in the Kantorovich Problem and Gromov-Wasserstein model to impose the guidance of keypoints in them. Meanwhile, the proposed KPG-RL model is extended to partial OT setting. As an application, we apply the proposed KPG-RL model to the heterogeneous domain adaptation. Experiments verified the effectiveness of the KPG-RL model. Code is available at https://github.com/XJTU-XGU/KPG-RL.

## 1 Introduction

As a mathematical tool for distribution alignment, mass transport, *etc.*, Optimal Transport (OT) [1] has gained increasing attention in the machine learning community. OT aims to derive a transport map or plan between a source and a target distribution, such that the transport cost is minimized. Due to its capacity to exploit the geometric property of data, OT has been employed in many applications, *e.g.*, computer vision [2, 3], natural language processing [4, 5, 6, 7], generative adversarial network [8], domain adaptation [9], clustering [10], anomaly detection [11], *etc*. OT has two typical formulations, *i.e.*, the Monge's formulation [12] and the Kantorovich's formulation [13]. The Kantorovich's formulation [13] relaxes the Monge's formulation and attracts broader studies in applications. Unless otherwise stated, by OT, we refer to the Kantorovich's formulation in this paper. The original OT model, *i.e.*, the Kantorovich Problem [13] (KP), is a linear program that is computationally expensive. The entropy-regularized OT [14] introduces the entropy of the transport plan as regularization to the OT model which is solved by the computationally cheaper Sinkhorn-Knopp algorithm [15] allowing to use automatic differentiation [16].

KP needs to transport all the mass of source distribution to exactly match the mass of target distribution. But in some cases, only partial mass of source and target distributions should be matched, *e.g.*, the source or target samples contain outliers. To overcome this limitation, the partial OT [17, 18, 19, 20],

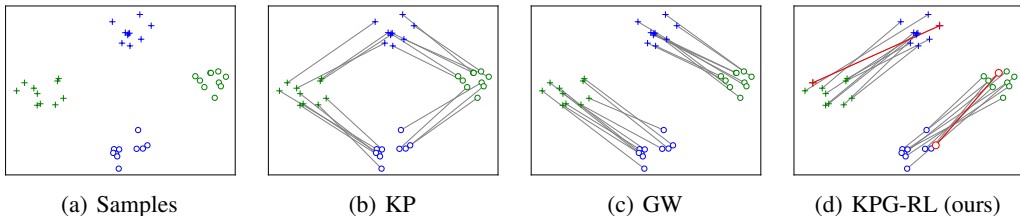

|     |     |     |     |
| :-: | :-: | :-: | :-: |
| (a) Samples | (b) KP | (c) GW | (d) KPG-RL (ours) |

Figure 1: (a) Positive (cross) and negative (circle) samples of source (in blue) and target (in green) distributions. (b) KP distorts the data structure, leading to mismatch of some positive and negative samples. (c) GW model better preserves the data structure, but completely mismatches the samples. (d) Our proposed KPG-RL model utilizes some keypoints (red pairs) to guide the transport and produce correct matching.

unbalanced OT [21, 22], and robust OT [23, 24, 25] models are presented, that allow to transport only partial mass of distribution. Another extension of KP is the Gromov-Wasserstein (GW) model [26, 27] that computes the distance between metrics defined within each domain rather than between samples across domains as in KP.

In most of the above models, the main criterion for the optimal transport plan/matching is by the minimization of total transport distance or distortion over all samples. Though having achieved promising results in many applications, without any additional guidance, the OT models may lead to incorrect matching of samples. Figures 1(b) and 1(c) illustrate examples of incorrect matching produced by KP and GW models. In many applications, it is reasonable to annotate some paired keypoints across domains for guiding the matching in OT. For instance, in non-rigid point set or image registration [28, 29], a few keypoint pairs that should be matched are annotated in two point sets/images. In semi-supervised domain adaptation [30] and heterogeneous domain adaptation [31], there are a few labeled target domain data and large amount of labeled source domain data available, that could be directly taken as keypoints. Therefore, it is valuable and important to investigate how to take advantage of those keypoints to guide the correct matching in OT. Figure 1(d) shows an example that with the guidance of a few keypoints, the correctness of matching can be improved.

In this paper, we propose a novel KeyPoint-Guided model by ReLation preservation (KPG-RL) for leveraging the annotated keypoints to guide the matching in OT. In KPG-RL, we first preserve the matching of keypoint pairs in OT using a mask-based constraint of the transport plan. We then propose to preserve the relation of each data point to the keypoints in transport, enforcing the matching of the data points near paired keypoints across domains. The proposed KPG-RL model is applicable even when distributions lie in different spaces, and can be solved by Sinkhorn's algorithm. We further enforce the relation preservation constraint in KP and GW to impose the guidance of keypoints in them. To tackle the problems that only partial mass should be transported, we extend the KPG-RL model to partial OT setting, forming partial-KPG-RL model.

As an application, we apply the KPG-RL model to the heterogeneous domain adaptation (HDA) [31]. HDA is a transfer learning task that aims to transfer the knowledge of large-scale labeled source domain data to the target domain where a few labeled and larger amounts of unlabeled data are available for training. The "heterogeneity" implies that the source and target domain data are in heterogeneous feature spaces, *e.g.*, generated by different deep networks. This heterogeneity poses a major obstacle in adapting the source trained model to the target domain. We take the labeled target domain data and source class centers as keypoints and transport source domain data to target domain by our KPG-RL model. Upon the transported source domain data and the labeled target domain data, a classification model is trained that is transferable to the target domain. Experiments show that the KPG-RL model is effective for HDA.

In the following sections, we discuss the related works in Sect. 2, and introduce the background of OT in Sect. 3. In Sect. 4, we discuss the details of the proposed KPG-RL model. In Sect. 5, we apply the KPG-RL model to HDA. Section 6 concludes this paper.

**Notations.** $\Sigma_m = \{\boldsymbol{p} \in \mathbb{R}_+^m | \sum_i p_i = 1\}$ is the probability simplex. $\langle \cdot, \cdot \rangle_F$ is the Frobenius dot product of two matrices. $\odot$ stands for the Hadamard product. $\mathbb{1}_m$ denotes $m$-dimensional all-one vector. $\pi_{i,:}$ and $\pi_{:,j}$ are respectively the $i$-th row and $j$-th column of matrix $\pi$.

## 2   Related Works

We review below the most related OT models and HDA methods to our work.

**OT models.** The GW model [27] seeks a "distance-preserving" transport plan such that the distance between transported points in target domain is the same as the distance between the original points in source domain. Our KPG-RL model aims to use keypoint pairs to guide the matching in OT by preserving the relation of each point to the keypoints. Our "relation-preserving" scheme preserves the relation of data w.r.t. the given keypoints, different from the pairwise distance-preserving constraint in GW. We experimentally verified the effectiveness of relation-preserving scheme for introducing the guidance of keypoints in OT. From the computational point of view, GW is a non-convex quadratic program, while KPG-RL is a linear program. Lin *et al.* [32] use the anchors to encourage clustering of data and to impose rank constraints on the transport plan to improve its robustness to outliers. The "anchors" in [32] are intermediate points in computation for improving robustness, different from the "keypoints" in this paper, which are the annotated paired data for guiding the matching in OT. Hierarchical OT [7, 33, 34] transports points by dividing them into some subgroups and then derives the transportation of these subgroups using OT. Different in goal and methodology from Hierarchical OT, we impose the guidance of keypoints for pursuing correct matching in OT by preserving the relation to the keypoints. We do not explicitly divide the points into subgroups, and there is no hierarchy in our method. TLB [27, 35] is a lower bound of GW that can be computed faster. TLB takes the ordered distance of each point to all the points in the same domain as features, and then performs the Kantorovich formulation of OT using such features. Differently, our method uses a carefully designed relation of each point to the keypoints to impose the guidance of keypoints to the other points. Courty et al. [9] constrain the cost function to encourage the matching of labeled data across source and target domains that share the same class labels for domain adaptation. They use the Laplacian regularization to preserve the data structure. Differently, we explicitly model the guidance of keypoints matching to the other data points in our OT formulation. The matching of paired keypoints is enforced by our mask-based constraint on the transport plan. Zhang et al. [36] propose Masked OT model as a regularization term to preserve the local feature invariances between fine-tuned and pretrained graph neural network (GNN) for the fine-tuning of GNNs. Though the Masked OT [36] shares similar spirits to the mask-based modeling in our method, our main contribution is the relation preservation for imposing the guidance of keypoints, different from [36] in methodology. For our mask-based modeling, it is utilized to impose the matching of keypoints which is theoretically guaranteed. While the mask in [36] aims to preserve the local information of finetuned network from pretrained models. The motivation and design of our mask are different from those in [36].

**HDA methods.** HDA methods could be roughly categorized into cross-domain mapping and common subspace learning methods. The cross-domain mapping approaches [31, 37, 38, 39, 40] learn a transform to map the source features or model parameters to target domain to achieve adaptation. The common subspace learning approaches [41, 42, 43, 44, 45, 46] learn domain-specific projections to map source and target domain data into a common subspace such that their distributions are aligned. Our method for HDA belongs to the first category, and could be mostly related to [37]. The method in [37] transports source samples to target domain using GW model regularized by the distance between the center of transported source samples and the center of labeled target samples having the same class labels. Different from [37], we take each labeled target domain data and its corresponding source domain class center as a paired keypoint and preserve the relation of each data to the set of keypoints in each domain when conducting OT.

## 3   Background on Optimal Transport

**Kantorovich Problem (KP).** We consider two sets of data points, *i.e.*, the source data $\boldsymbol{X} = \{x_i\}_{i=1}^m$ and the target data $\boldsymbol{Y} = \{y_j\}_{j=1}^n$, of which the empirical distributions are $\boldsymbol{p} = \sum_{i=1}^m p_i \delta_{x_i}$ and $\boldsymbol{q} = \sum_{j=1}^n q_j \delta_{y_j}$. With a slight abuse of notations, we may also denote $\boldsymbol{p} = (p_1, p_2, \cdots, p_m)^\top \in \Sigma_m$ and $\boldsymbol{q} = (q_1, q_2, \cdots, q_n)^\top \in \Sigma_n$ as the mass supported on $\boldsymbol{X}$ and $\boldsymbol{Y}$ respectively. We define the cost matrix between $\boldsymbol{X}$ and $\boldsymbol{Y}$ as $C = (C_{i,j}) \in \mathbb{R}^{m \times n}$ with $C_{i,j} = c(x_i, y_j)$, where $c$ is a cost function, which is set to the squared $L_2$-distance of $x_i$ and $y_j$ in our experiments. OT aims to optimally transport $\boldsymbol{p}$ towards $\boldsymbol{q}$ at the smallest cost, formulated as the following Kantorovich Problem (KP):

$$\min_{\pi \in \Pi(\boldsymbol{p}, \boldsymbol{q})} L_{kp}(\pi) \triangleq \langle \pi, C \rangle_F, \text{ s.t. } \Pi(\boldsymbol{p}, \boldsymbol{q}) = \{\pi \in \mathbb{R}_+^{m \times n} | \pi \mathbb{1}_n = \boldsymbol{p}, \pi^\top \mathbb{1}_m = \boldsymbol{q}\}. \qquad (1)$$

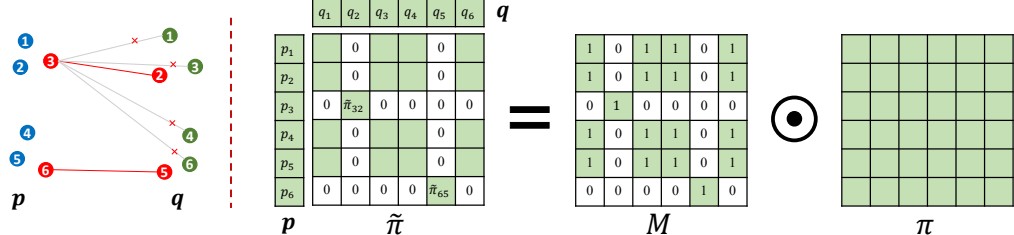

Figure 2: Example of modeling the matching of keypoints (red) using mask. The left part illustrates the matching of data points with keypoint pairs $\mathcal{K} = \{(3,2), (6,5)\}$. To preserve the matching of keypoint index pair $(i, j)$, $e.g.$, $(3, 2)$, the $i$-th row and $j$-th column of transport plan $\tilde{\pi}$ must be zeros except $\tilde{\pi}_{i,j}$. Therefore, we can model $\tilde{\pi}$ by $\tilde{\pi} = M \odot \pi$, where $M$ is the mask determined by the keypoints, and $\pi$ is to be optimized.

When $c$ is taken as a distance metric (*aka.* ground metric), the minimum value of objective function in Eq. (1) is a distance between $p$ and $q$, named Wasserstein distance.

**Partial OT model.** The KP in Eq. (1) takes the mass preserving assumption that all the mass of $p$ is transported to exactly match the mass of $q$. In many applications, only partial mass should be transported. The partial OT model [18, 19] seeks the minimal cost of transporting only $s$ unit mass from $p$ to $q$, where $0 \leqslant s \leqslant \min(\|p\|_1, \|q\|_1)$, formulated as

$$\min_{\pi \in \Pi^s(p,q)} L_{kp}(\pi), \text{ s.t. } \Pi^s(p, q) = \{\pi \in \mathbb{R}_+^{m \times n} | \pi \mathbb{1}_n \leqslant p, \pi^\top \mathbb{1}_m \leqslant q, \mathbb{1}_m^\top \pi \mathbb{1}_n = s\}. \quad (2)$$

Note that in partial OT, the total mass $\|p\|_1$ and $\|q\|_1$ are not necessarily equal. In the definition of $\Pi(p, q)$ and $\Pi^s(p, q)$, $m$ and $n$ are respectively the length of $p$ and $q$.

**Gromov-Wasserstein (GW) model.** If the data points $x_i$ and $y_j$ lie in different spaces, the distance between $x_i$ and $y_j$ may not be computed. The GW [27] model then minimizes the distortion when transporting the whole set of points from one space to another. The GW model relies on the intra-domain distance of source domain as $C^s = (C_{i,k}^s) \in \mathbb{R}^{m \times m}$ and target domain as $C^t = (C_{j,l}^t) \in \mathbb{R}^{n \times n}$ where $C_{i,k}^s$ is the distance between source domain data $x_i$ and $x_k$, and $C_{j,l}^t$ is the distance between target domain data $y_j$ and $y_l$. The GW model is given by

$$\min_{\pi \in \Pi(p,q)} L_{gw}(\pi) \triangleq \sum_{i,k=1}^m \sum_{j,l=1}^n \pi_{i,j} \pi_{k,l} |C_{i,k}^s - C_{j,l}^t|^2. \quad (3)$$

## 4 Keypoint-Guided Optimal Transport

This section details our proposed keypoint-guided model that leverages the keypoints to guide the matching in OT. The guidance is imposed by preserving the matching of keypoint pairs and the relation of each data point to the keypoints. We first discuss the preservation of matching of keypoints, then introduce the modeling of relation, and finally present the keypoint-guided OT models.

**Preservation of matching of keypoints in transport.** We denote the set of keypoint index pairs as $\mathcal{K} = \{(i_u, j_u)\}_{u=1}^U$ with $U$ denoting the number of paired keypoints. We respectively denote $\mathcal{I} = \{i_u\}_{u=1}^U$ and $\mathcal{J} = \{j_u\}_{u=1}^U$ as the sets of source and target keypoint indexes. For the example illustrated in Fig. 2, $\mathcal{K} = \{(3, 2), (6, 5)\}$, $\mathcal{I} = \{3, 6\}$, and $\mathcal{J} = \{2, 5\}$. To impose the guidance of these keypoints with indexes in $\mathcal{K}$ in deriving the matching in OT, we first guarantee the matching of the keypoint pairs. As illustrated in Fig. 2, we preserve the matching of keypoints in transport using a mask-based constraint of the transport plan, which is motivated by the following observation. If the paired keypoints $(i, j) \in \mathcal{K}$ are matched, the optimal transport plan $\tilde{\pi}$ satisfies that the $i$-th row and $j$-th column of $\tilde{\pi}$ must be zeros except $\tilde{\pi}_{i,j}$, which means that the all mass of source keypoint $x_i$ must be transported to target keypoint $y_j$ and $y_j$ can only receive the mass from $x_i$. For the example in Fig. 2, the 3-th row and 2-th column are zeros except that $\tilde{\pi}_{3,2} > 0$. This sparsity of $\tilde{\pi}$ motivates us to model it as the Hadamard product of a mask matrix $M = (M_{i,j}) \in \mathbb{R}^{m \times n}$ and a matrix $\pi \in \mathbb{R}_+^{m \times n}$ with positive entries, *i.e.*,

$$\tilde{\pi} = M \odot \pi, \text{ with } \tilde{\pi}_{i,j} = M_{i,j} \pi_{i,j}. \quad (4)$$

With Eq. (4), we define the admissible solution set for our keypoint-guided OT model as

$$\Pi(\boldsymbol{p}, \boldsymbol{q}; M) = \{\pi \in \mathbb{R}_+^{m \times n} | (M \odot \pi)\mathbb{1}_n = \boldsymbol{p}, (M \odot \pi)^\top \mathbb{1}_m = \boldsymbol{q}\}. \qquad (5)$$

Note that the $\pi$ in Eq. (5) is not necessarily a coupling. The entry $M_{i,j}$ of the mask matrix $M$ is set to 0 if $\tilde{\pi}_{i,j}$ needs to be 0, otherwise $M_{i,j}$ is set to 1. Figure 2 illustrates the mask-based modeling of Eq. (4). $M$ is constructed as in Proposition 1.

**Proposition 1.** *Suppose that the mask matrix $M$ satisfies that*

$$M_{i,j} = \begin{cases} 1, & \text{if } (i,j) \in \mathcal{K}, \\ 0, & \text{if } i \in \mathcal{I} \text{ and } (i,j) \notin \mathcal{K}, \\ 0, & \text{if } j \in \mathcal{J} \text{ and } (i,j) \notin \mathcal{K}, \\ 1, & \text{otherwise (i.e., } i \notin \mathcal{I} \text{ and } j \notin \mathcal{J}). \end{cases} \qquad (6)$$

*and $p_i = q_j$, for $(i,j) \in \mathcal{K}$. Then, the transport plan $\tilde{\pi} = M \odot \pi$ with $\pi \in \Pi(\boldsymbol{p}, \boldsymbol{q}; M)$ preserves the matching of paired keypoints with index pairs in $\mathcal{K}$.*

More explanations of the construction of $M$ and the proof of Proposition 1 are provided in Appendix A.1. Proposition 1 indicates that if $p_i = q_j$, for $(i,j) \in \mathcal{K}$, the matching of keypoint pairs is preserved by the mask-based constraint. This mask-based modeling in Eq. (4) is also applicable even for the case that there exist some $(i,j) \in \mathcal{K}$ such that $p_i \neq q_j$. For this case, we shall use different mask matrices. Please refer to Appendix A.1 for the details.

**Modeling the relation to keypoints.** To use the keypoints to guide the matching in transport, we propose to preserve the relation of each point to the set of keypoints in transport. Figure 3 illustrates the relation within points of each distribution. For the data point $x_k \in \boldsymbol{X}$, its relation score to the keypoint $x_{i_u}$ for $i_u \in \mathcal{I}$ (illustrated by red circle in the left of Fig. 3) is defined as

$$R_{k,i_u}^s = \frac{e^{-C_{k,i_u}^s/\tau}}{\sum_{u'=1}^{U} e^{-C_{k,i_u}^s/\tau}}, \ \forall i_u \in \mathcal{I}, \quad (7)$$

where $C_{k,i_u}^s$ is the $l_2$-distance between $x_k$ and $x_{i_u}$, and $\tau$ is temperature set as $\tau = \rho * \max_{i,k}\{C_{i,k}^s\}$ in experiments. Similarly, for $y_l \in \boldsymbol{Y}$, its relation score to the keypoint $y_{j_u}$ for $j_u \in \mathcal{J}$ is defined as

$$R_{l,j_u}^t = \frac{e^{-C_{l,j_u}^t/\tau'}}{\sum_{u'=1}^{U} e^{-C_{l,j_u}^t/\tau'}}, \ \forall j_u \in \mathcal{J}, \quad (8)$$

where $C_{l,j_u}^t$ is the distance between $y_l$ and $y_{j_u}$, $\tau'$ is temperature set to $\rho * \max_{j,l}\{C_{j,l}^t\}$. The setting of $\tau$ and $\tau'$ implies that the distances

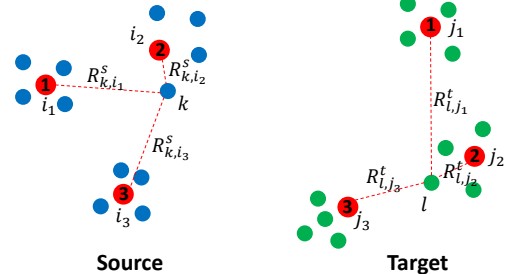

**Source** **Target**

Figure 3: Illustration of relation of each point to the keypoints (red). In the keypoint-guided OT model, the relation of each point to keypoints should be preserved after transport.

are divided by their maximum value, and normalized to $[0, 1]$, which increases the robustness of the relation score to the scale of the distances. $\rho$ is a tunable parameter and set to 0.1 in our experiments, since 0.1 is a commonly used temperature in the softmax function [47, 48]. We denote $R_k^s = (R_{k,i_1}^s, R_{k,i_2}^s, \cdots, R_{k,i_U}^s)$ and $R_l^t = (R_{l,j_1}^t, R_{l,j_2}^t, \cdots, R_{l,j_U}^t)$ that represent the relation of data points $x_k$ and $y_l$ to the keypoints in source and target domains respectively. From the definition of the relation, the cross domain points near to a paired keypoints share similar relation to the keypoints in corresponding domain. For instance, in Fig. 3, $x_k$ and $y_l$ near to the paired keypoints with indexes $(i_2, j_2)$ have similar relation $R_k^s$ and $R_l^t$. Meanwhile, if $x_k$ is distant from all the keypoints, $R_k^s$ is close to a uniform probability vector.

**Keypoint-guided model.** Based on the relation given above, we define the guiding matrix $G = (G_{k,l}) \in \mathbb{R}^{m \times n}$ by $G_{k,l} = d(R_k^s, R_l^t)$ where $d$ measures the dissimilarity of $R_k^s$ and $R_l^t$. Then, $G_{k,l}$ is with small and even near to 0 if $R_k^s$ and $R_l^t$ are similar. $d$ is taken as the Jensen–Shannon divergence in this paper. We will study the effect of $d$ in Appendix B.5. By using the mask-based constraint of transport plan, the *KeyPoint-Guided model by ReLation preservation* (**KPG-RL**) is defined as

$$\min_{\pi \in \Pi(\boldsymbol{p}, \boldsymbol{q}; M)} L_{kpg}(\pi) \triangleq \langle M \odot \pi, G \rangle_F. \qquad (9)$$

By the KPG-RL model in Eq. (9), first, the matching of keypoint pairs is enforced by the mask-based constraint of the transport plan. Second, the minimization of the objective function enforces that the optimal transport plan has larger entries in the locations where the entries of $G$ are smaller. Hence the cross-domain points corresponding to these locations (*e.g.*, $k$ and $l$ shown in Fig. 3) that are near to a paired keypoints tend to be matched. Based on the softmax-based formulations in Eqs. (7) and (8), $d(R_k^s, R_l^t)$ is mainly determined by the relation score to the closest keypoint(s), since relation scores to the distant keypoints are small or close to 0. This implies that the points are mainly guided by the closest keypoints in our KPG-RL model in Eq. (9). For the points distant from all the keypoints, their corresponding entries of $G$ are close to zeros according to the definition of $G$. Therefore, the guidance of keypoints to these points is limited. To achieve correct matching of these points, additional information, *e.g.*, point-wise cost or more keypoints, is needed.

Equation (9) is a linear program and can be solved by linear programming algorithms, *e.g.*, the Simplex algorithm. We give the details for reformulating Eq. (9) as the standard form of linear programming in Appendix A.2. Since Sinkhorn's algorithm offers a lightspeed computation of the entropy-regularized OT [14], a natural question is that, can Sinkhorn's algorithm be applied to the KPG-RL model with entropy regularization? We give the positive answer, and the details of the deduction are given in Appendix A.3. The iterative formulas are

$$\boldsymbol{u}^{(l+1)} = \frac{\boldsymbol{p}}{K\boldsymbol{v}^l}, \quad \boldsymbol{v}^{(l+1)} = \frac{\boldsymbol{q}}{K^\top \boldsymbol{u}^{(l+1)}}, \tag{10}$$

where $K = M \odot e^{-G/\epsilon}$, $\epsilon$ is the coefficient of entropy regularization. The division operator used above is entry-wise. After iteration, the optimal transport plan is $\mathrm{diag}(\boldsymbol{u})K\mathrm{diag}(\boldsymbol{v})$.

Note that in this KPG-RL model in Eq. (9), the points in $\boldsymbol{X}$ and $\boldsymbol{Y}$ do not necessarily lie in the same space because we only need to compute the distance within each one of $\boldsymbol{X}$ and $\boldsymbol{Y}$. Therefore the proposed KPG-RL model in Eq. (9) is applicable even when $\boldsymbol{p}$ and $\boldsymbol{q}$ are supported in different spaces. As mentioned in Sect. 1, the GW model is applicable for transport across different spaces. However, GW is a non-convex quadratic program and is often solved by the Frank-Walfe algorithm in which a KP-like problem needs to be solved at each iteration by Sinkhorn's algorithm or linear programming. Surprisingly, our KPG-RL model can be directly solved using Sinkhorn's algorithm or linear programming without additional iteration, as discussed above.

**Imposing keypoint guidance in KP/GW.** We can impose the keypoint guidance in KP by adding $L_{kpg}(\pi)$ as a regularization term to KP, obtaining the following **KPG-RL-KP** model:

$$\min_{\pi \in \Pi(\boldsymbol{p},\boldsymbol{q};M)} \left\{ \alpha L_{kp}(M \odot \pi) + (1-\alpha)L_{kpg}(\pi) = \langle M \odot \pi, \alpha C + (1-\alpha)G \rangle_F \right\}, \alpha \in (0,1). \tag{11}$$

The KPG-RL-KP can be solved by Sinkhorn's algorithm or linear programming, the same as the solution of KPG-RL model. Similarly, we define the **KPG-RL-GW** model in GW by

$$\min_{\pi \in \Pi(\boldsymbol{p},\boldsymbol{q};M)} \alpha L_{gw}(M \odot \pi) + (1-\alpha)L_{kpg}(\pi), \alpha \in (0,1). \tag{12}$$

The KPG-RL-GW model is solved using the Frank-Walfe algorithm of which the details are given in Appendix A.4. The KPG-RL-KP/KPG-RL-GW models take both advantages of KP/GW and KPG-RL models, and could be helpful especially when the number of paired keypoints is small. We show in Appendix A.5 that the KPG-RL-KP and KPG-RL-GW provde a proper metric and a divergence under mild conditions. $\alpha$ is simply set to 0.5 in our experiments.

**Extension to partial OT.** We now extend the above KPG-RL model to the more practical partial OT setting that only partial mass is transported. To do this, we first apply the above mask-based constraint of transport plan to the partial OT model in Eq. (2). We then add more constraints to enforce that all the mass of keypoints is transported to preserve the matching of keypoints. Finally, we define the **partial-KPG-RL** model as

$$\min_{\pi \in \Pi^s(\boldsymbol{p},\boldsymbol{q};M)} \left\{ L_{kpg}(M \odot \pi) = \langle M \odot \pi, G \rangle_F \right\}, \tag{13}$$

where $\Pi^s(\boldsymbol{p}, \boldsymbol{q}; M) = \{\pi \in \mathbb{R}_+^{m \times n} | (M \odot \pi)\mathbb{1}_n \leqslant \boldsymbol{p}, (M \odot \pi)^\top \mathbb{1}_m \leqslant \boldsymbol{q}, \mathbb{1}_m^\top (M \odot \pi)\mathbb{1}_n = s; (M \odot \pi)_{i,:}\mathbb{1}_n = p_i, \forall i \in \mathcal{I}; \mathbb{1}_m^\top (M \odot \pi)_{:,j} = q_j, \forall j \in \mathcal{J}\}$. Chapel *et al.* [20] propose a compelling method to solve the original partial OT problem in Eq. (2) by transforming the partial OT problem into an OT-like problem. Inspired by Chapel *et al.* [20], to solve our partial-KPG-RL model

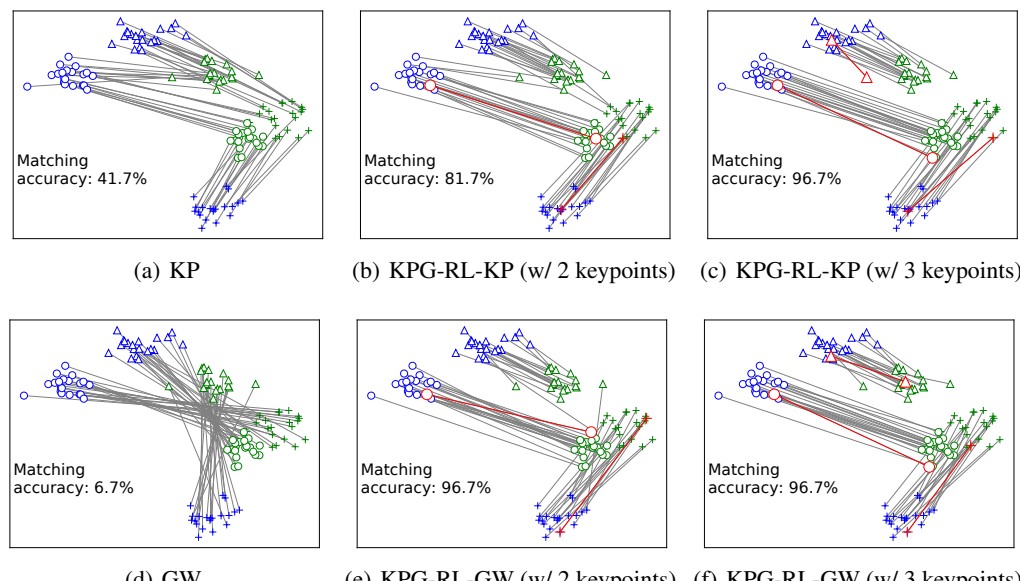

Figure 4: Matching produced by (a) KP, (b) KPG-RL-KP model given 2 keypoint pairs, (c) KPG-RL-KP model given 3 keypoint pairs, (d) GW model, (e) KPG-RL-GW model given 2 keypoint pairs, and (f) KPG-RL-GW model given 3 keypoint pairs.

in Eq. (13), we add a dummy point to each of the marginal distributions to transform Eq. (13) into a KPG-RL-like problem. To do this, we extend $\boldsymbol{p}, \boldsymbol{q}, G, M$ by

$$\bar{\boldsymbol{p}} = \begin{bmatrix} \boldsymbol{p} \\ \|\boldsymbol{q}\|_1 - s \end{bmatrix}, \bar{\boldsymbol{q}} = \begin{bmatrix} \boldsymbol{q} \\ \|\boldsymbol{p}\|_1 - s \end{bmatrix}, \bar{G} = \begin{bmatrix} G & \xi\mathbb{1}_n \\ \xi\mathbb{1}_m^\top & 2\xi + A \end{bmatrix}, \bar{M} = \begin{bmatrix} M & \boldsymbol{a} \\ \boldsymbol{b}^\top & 1 \end{bmatrix},$$

where $A > 0, \xi > 0, \boldsymbol{a} \in \mathbb{R}^m, \boldsymbol{b} \in \mathbb{R}^n$. The element $a_i$ of $\boldsymbol{a}$ is 0 if $i \in \mathcal{I}$, otherwise 1. The element $b_j$ of $\boldsymbol{b}$ is 0 if $j \in \mathcal{J}$, and 1 otherwise. For more motivations of this extension, please refer to Appendix A.6. By the following theorem, solving the optimal transport plan of problem (13) boils down to solving the problem $\min_{\bar{\pi} \in \Pi(\bar{\boldsymbol{p}}, \bar{\boldsymbol{q}}; \bar{M})} \langle \bar{M} \odot \bar{\pi}, \bar{G} \rangle_F$.

**Theorem 1.** *Suppose $A > 0$, $\xi > 0$, $\sum_{i \in I} p_i < s$, and $\sum_{j \in J} q_j < s$, then the optimal transport plan $M \odot \pi^*$ of problem (13) is the $m$-by-$n$ block in the upper left corner of the optimal transport plan $\bar{M} \odot \bar{\pi}^*$ of problem $\min_{\bar{\pi} \in \Pi(\bar{\boldsymbol{p}}, \bar{\boldsymbol{q}}; \bar{M})} \langle \bar{M} \odot \bar{\pi}, \bar{G} \rangle_F$.*

Please refer to Appendix A.7 for the proof of Theorem 1.

## 5 Experiments

We evaluate the keypoint-guided models on toy data, and apply them to HDA and open-set HDA.

### 5.1 Toy Data

As illustrated in Fig. 4, in this toy data experiment, each of the source (blue) and target (green) distributions is a Gaussian mixture composed of three distinct Gaussian components indicated by different shapes where the same shapes indicate points of the same class. In Fig. 4, we have the following observations. In Fig. 4(a), in the KP model, the points in each component of target distribution are mismatched to points in different classes of source distributions, and only a small fraction of target points are correctly matched to source points belonging to the same class. In Fig. 4(b), given 2 keypoint pairs (from distinct classes), the KPG-RL-KP model apparently improves the correctness of matching. In Fig. 4(c), the KPG-RL-KP model mainly matches the source points to the target points belonging to the same class, thanks to the given 3 keypoint pairs. This implies that our proposed keypoint-guided model does help to produce correct matching in OT by leveraging the

Table 1: Accuracy on Office-31 for HDA. "A", "W", and "D" are respectively the domains of amazon, webcam, and dslr. "· → ∗" denotes a heterogeneous adaptation task where · and ∗ are respectively source domain using DeCAF$_6$ feature and target domain using ResNet-50 feature.

| Method | A→A | A→D | A→W | D→A | D→D | D→W | W→A | W→D | W→W | **Avg** |
|---|---|---|---|---|---|---|---|---|---|---|
| Labeled-target-only | 45.3 | 69.2 | 67.3 | 45.3 | 69.2 | 67.3 | 45.3 | 69.2 | 67.3 | 60.6 |
| STN [50] | 58.7 | 84.8 | 80.0 | 51.2 | 91.0 | 83.8 | 52.8 | 95.2 | 87.4 | 76.1 |
| SSAN [51] | 56.8 | 89.7 | **87.1** | 54.2 | 82.6 | **90.3** | 50.0 | 85.8 | 85.8 | 75.8 |
| DDACL [52] | 44.2 | 63.2 | 64.2 | 43.9 | 77.4 | 70.6 | 39.8 | 64.5 | 73.2 | 60.1 |
| CDSPP [45] | 55.5 | 79.7 | 76.5 | 43.2 | 80.6 | 84.2 | 47.4 | 78.7 | 84.5 | 70.0 |
| SGW [37] | 49.7 | 77.7 | 73.6 | 49.4 | 78.4 | 73.6 | 48.7 | 80.3 | 74.5 | 67.3 |
| GW [27] | 33.6 | 41.6 | 35.5 | 39.7 | 40.0 | 31.7 | 34.8 | 34.5 | 29.0 | 35.6 |
| **GW (w/ mask)** | 41.3 | 71.6 | 69.7 | 41.9 | 71.2 | 69.8 | 40.3 | 71.6 | 69.7 | 60.8 |
| HOT [33] | 39.0 | 44.8 | 40.0 | 31.3 | 52.6 | 44.8 | 29.7 | 60.0 | 56.5 | 44.3 |
| **HOT (w/ mask)** | 45.2 | 60.3 | 57.4 | 48.9 | 63.5 | 59.2 | 40.3 | 67.1 | 61.4 | 55.9 |
| TLB [27] | 29.4 | 36.5 | 43.2 | 24.5 | 31.3 | 51.0 | 23.6 | 31.9 | 49.7 | 35.7 |
| **TLB (w/ mask)** | 42.5 | 66.3 | 64.7 | 38.5 | 68.5 | 65.9 | 43.1 | 68.2 | 67.3 | 58.3 |
| **KPG (w/ dist)** | 55.2 | 60.7 | 71.6 | 51.3 | 71.9 | 77.1 | 48.7 | 70.0 | 77.7 | 64.9 |
| **KPG-RL-GW** | 58.7 | 92.9 | 84.2 | 57.4 | 95.5 | 87.1 | 55.5 | **95.5** | **90.0** | 79.6 |
| **KPG-RL (LP)** | 56.5 | **93.6** | 83.2 | **58.1** | 94.5 | 86.8 | 55.8 | 95.2 | 89.7 | 79.3 |
| **KPG-RL (SH)** | **60.0** | 91.6 | 83.6 | 57.4 | **95.8** | 87.7 | **59.1** | 95.2 | 88.4 | **79.9** |

given a few keypoint pairs. In Figs. 4(d), 4(e) and 4(f), the proposed KPG-RL-GW model improves the correctness of matching of GW model by leveraging the guidance of given keypoints pairs. Please refer to Appendix B.1 for the toy experiment for evaluating the partial-KPG-RL model.

## 5.2 Heterogeneous Domain Adaptation

In HDA, we are given a large amount of labeled source domain data $\{(x_i, t_i)\}_{i=1}^m$, a few labeled target domain data $\{y_j, \bar{t}_j\}_{j=1}^{n_l}$, and large number of unlabeled target domain data $\{y_j\}_{j=n_l+1}^n$, where $m \gg n_l$, $n \gg n_l$, $x_i$ and $y_j$ are features, and $t_i$ and $\bar{t}_j$ are respectively the class labels of $x_i$ and $y_j$. The heterogeneity means that $x_i$ and $y_j$ are from different spaces/modalities. The goal of HDA is to train a classification model using the given data to predict the label of unlabeled target domain data, leveraging the knowledge of large-scale labeled source domain data. The main challenge is that the domain gap between heterogeneous source and target distributions supported in distinct spaces hinders the direct employment of the source domain trained model in the target domain.

We tackle the problem of HDA using our proposed KPG-RL model as follows. We first transport the source domain data using our KPG-RL model to the target domain. More concretely, in KPG-RL, each labeled target domain data and the source domain class center of the same class are taken as a matched keypoint pair. The KPG-RL model is then performed between empirical distributions of the source domain data along with class centers and the target domain data (including labeled and unlabeled data). Based on the produced optimal transport plan, we transport the source domain data using the barycentric mapping [49] to the target domain. Finally, we train the classification model (taken as a kernel SVM) on the transported source domain data (using their class label before transport) and the labeled target domain data, which is applied to the target domain test data. $\epsilon$ is set to 0.005. More details of the barycentric mapping, the kernel SVM, *etc*, are given in Appendix B.4.

We compare our method with the following baseline methods, including 1) "Labeled-target-only" that trains the kernel SVM using the labeled target data; 2) the OT methods of "GW", "HOT", and "TLB" that transport source domain data using barycentric mapping induced by the transport plan of GW [27], Hierarchical OT [33], and TLB [27], and then train the kernel SVM on the transported data and labeled target domain data; 3) the typical HDA methods of "SGW" [37] and "STN" [50], and the recent HDA methods of "SSAN" [51], "DDACL" [52] and "CDSPP" [45]. We conduct experiments on Office-31 [53] dataset. On Office-31, we use the DeCAF$_6$ [54] features and the features extracted by ResNet-50 [55] pretrained on ImageNet [56] to respectively build source and target domains for constructing heterogeneous adaptation tasks. In each task, one labeled data (1-shot) for each class is

Table 2: Results of methods of Labeled-target-only and KPG given different shots (1, 2, and 3) of labeled target domain data (keypoints) per-class under five distinct samplings (S1,S2,S3,S4, and S5).

| Method | 1-shot | | | | | | 2-shot | | | | | | 3-shot | | | | | |
| --- | --- | --- | --- | --- | --- | --- | --- | --- | --- | --- | --- | --- | --- | --- | --- | --- | --- | --- |
| | S1 | S2 | S3 | S4 | S5 | Avg | S1 | S2 | S3 | S4 | S5 | Avg | S1 | S2 | S3 | S4 | S5 | Avg |
| Labeled-target-only | 61.2 | 58.4 | 64.1 | 60.6 | 58.7 | 60.6 | 77.8 | 76.3 | 78.3 | 75.0 | 74.2 | 76.3 | 80.7 | 76.6 | 81.0 | 83.4 | 82.7 | 80.9 |
| **KPG-RL (SH)** | **77.6** | **76.2** | **82.1** | **82.9** | **80.7** | **79.9** | **86.5** | **86.5** | **86.8** | **84.4** | **82.6** | **85.4** | **86.5** | **88.5** | **87.0** | **88.3** | **88.2** | **87.7** |

given in target domain. Note all the methods use the same training data (including labeled source and target domain data, and unlabeled target domain data) and test data.

Table 1 reports the results on Offce-31 dataset. In Table 1, "KPG-RL (SH)" and "KPG-RL (LP)" denote our KPG-RL model solved using Sinkhorn's algorithm and linear programming respectively. KPG-RL (SH) achieves marginal better average accuracy than KPG-RL (LP), which could be because the barycentric mapping based on the more dense transport plan optimized by Sinkhorn's algorithm is more robust to wrong matching. We also observe that KPG-RL-GW achieves slightly lower average accuracy than KPG-RL (SH), and KPG-RL (SH) is more computationally efficient as discussed in Sect. 4. "KPG (w/ dist)" is the approach that imposes the guidance of keypoints using distance preservation in our framework, *i.e.*, $R_{k,i_u}^s$ and $R_{l,j_u}^t$ in Eqs. (7) and (8) are taken as $C_{k,i_u}^s$ and $C_{l,j_u}^t$ respectively. KPG-RL (SH) improves the accuracy of KPG (w/ dist) by 15%, implying that our relation-preserving scheme is more effective for imposing the guidance of keypoints than the distance-preserving scheme. Table 1 shows that the results of GW, HOT, and TLB are inferior to that of Labeled-target-only, indicating that GW, HOT, and TLB cause negative transfer. This is reasonable because GW, HOT, and TLB do not take advantage of the guidance of keypoints and can cause wrong matching. We also use our mask-based constraint on the transport plan in GW, HOT, and TLB, of which the corresponding approach are denoted as GW (w/ mask), HOT (w/ mask), TLB (w/ mask). We can observe that with the mask-based constraint, the performances of GW, HOT, and TLB are improved but worse than KPG-RL (SH) and KPG-RL (LP), verifying the importance of the relation preservation scheme in KPG-RL. SGW [37] aligns the class centers of labeled target domain data and transported source domain data in the GW model, and realizes positive transfer. Our proposed KPG-RL (SH) outperforms SGW by 12.2%, confirming that our KPG-RL model preserving relation is more effective than SGW preserving class centers for HDA. Compared with the other HDA methods, KPG-RL (SH) achieves the best average accuracy.

The keypoints are important to KPG-RL, since they are used to guide the matching. We show the results with varying keypoints by giving different numbers and sampling of labeled target domain data, in Table 2. It can be observed in Table 2 that under different numbers and samplings of labeled target domain data, our approach KPG-RL (SH) consistently outperforms the baseline Labeled-target-only, achieving positive transfer. We also find that as the number of labeled target domain data increases, the margin between the accuracy of Labeled-target-only and KPG-RL (SH) decreases. This indicates that given more labeled target domain data, the necessity of knowledge transferred from source domain becomes smaller. Please refer to Appendix B.5 for the effect of source keypoints.

### 5.3 Open-Set Heterogeneous Domain Adaptation

We conduct open-set HDA experiments to evaluate our proposed partial-KPG-RL model. The problem setting of open-set HDA is the same as HDA, except that the unlabeled target domain data contain samples of unknown classes absent in the categories of labeled data. The goal of open-set HDA is to correctly classify the common class samples and to detect the unknown class samples. We consider the case that the proportion $\eta$ of unknown class samples in the unlabeled target domain data is given. For applying partial-KPG-RL to open-set HDA, the keypoint pairs are constructed using labeled target domain data and their corresponding source class centers, same as KPG-RL for HDA in Sect. 5.2. We use the partial-KPG-RL model in Eq. (13) to transport all the labeled source domain data to partially match the target domain data. The unmatched target domain data are detected as unknown class. We then train a kernel SVM on the transported source domain data and labeled target domain data to classify the common class samples detected by partial-KPG-RL. The natural Baseline for open-set HDA is the approach that rejects the $\eta$-proportion unlabeled samples with largest distance to labeled target domain data as unknown, and trains a kernel SVM on the labeled target domain data to classify common class data. We also evaluate the HDA methods STN [50] and SSAN [51] in the

Table 3: Accuracy of common class sample (OS$^*$), Accuracy of unknown class sample (UNK), and their harmonic mean (HOS) on Office-31 for open-set HDA ($\eta = 0.67$).

| Method | A→A | | | A→D | | | A→W | | | D→A | | | D→D | | |
|---|---|---|---|---|---|---|---|---|---|---|---|---|---|---|---|
| | OS$^*$ | UNK | HOS | OS$^*$ | UNK | HOS | OS$^*$ | UNK | HOS | OS$^*$ | UNK | HOS | OS$^*$ | UNK | HOS |
| Baseline | 40.0 | 71.0 | 51.2 | 37.3 | 85.3 | 51.9 | 29.1 | 82.7 | 43.0 | 40.0 | 71.0 | 51.2 | 37.3 | 85.3 | 51.9 |
| STN [50] | **48.2** | **80.6** | **60.3** | **67.3** | **86.6** | **75.7** | 54.6 | 78.4 | 64.3 | 46.3 | 76.6 | 57.8 | 63.6 | 84.4 | 72.6 |
| SSAN [51] | 25.4 | 66.7 | 36.8 | 29.1 | 68.4 | 40.8 | **64.5** | **83.1** | **72.7** | 34.6 | 70.6 | 46.4 | 22.7 | 64.1 | 33.6 |
| **Partial-KPG-RL** | 41.8 | 77.1 | 54.2 | 48.2 | 74.9 | 58.6 | 38.2 | 73.2 | 50.2 | **52.7** | **82.3** | **64.3** | **80.0** | **92.6** | **85.9** |

| Method | D→W | | | W→A | | | W→D | | | W→W | | | Avg | | |
|---|---|---|---|---|---|---|---|---|---|---|---|---|---|---|---|
| | OS$^*$ | UNK | HOS | OS$^*$ | UNK | HOS | OS$^*$ | UNK | HOS | OS$^*$ | UNK | HOS | OS$^*$ | UNK | HOS |
| Baseline | 29.1 | 82.7 | 43.0 | 40.0 | 71.0 | 51.2 | 37.3 | 85.3 | 51.9 | 29.1 | 82.7 | 43.0 | 35.5 | 79.7 | 48.7 |
| STN [50] | 54.6 | 79.2 | 64.6 | 49.9 | 78.4 | 60.4 | 60.0 | 82.6 | 69.5 | 55.5 | 79.2 | 65.2 | 55.6 | 80.7 | 65.6 |
| SSAN [51] | 29.1 | 66.4 | 40.4 | 31.8 | 68.4 | 43.3 | 19.1 | 64.5 | 29.4 | 26.4 | 67.9 | 37.9 | 31.4 | 68.9 | 42.4 |
| **Partial-KPG-RL** | **73.6** | **89.2** | **80.7** | **52.7** | **82.7** | **64.4** | **78.2** | **90.9** | **84.1** | **71.8** | **88.3** | **79.2** | **59.7** | **83.5** | **69.1** |

open-set HDA task. For STN [50] and SSAN [51], we reject the $\eta$-proportion unlabeled samples with lowest prediction confidence as unknown class. Table 3 reports the results. We use the open-set evaluation metrics [57] including accuracy of common class sample (OS$^*$), accuracy of unknown class sample (UNK), and their harmonic mean HOS $= 2\frac{\text{OS}^* \times \text{UNK}}{\text{OS}^* + \text{UNK}}$. We can observe in Table 3 that our proposed partial-KPG-RL achieves the best results in terms of average OS$^*$, UNK, and HOS, indicating that the partial-KPG-RL is effective for both classifying common class data and identifying unknown class data. Compared with the Baseline, partial-KPG-RL outperforms it by 20.4% in terms of average HOS, confirming the positive transfer achieved by our method. More implementation details and the solution for Open-Set HDA with unknown $\eta$ are given in Appendix B.2.

Due to space limit, we include the experiment of our approach for deep unsupervised domain adaptation in Appendix B.3. More empirical analysis and ablation studies, *e.g.*, sensitivity to hyper-parameters, the effect of $d$, the discussion on how to define the keypoints in more general practical applications, the time and memory cost, *etc.*, are given in Appendix B.5.

## 6  Conclusion

In this paper, we propose a novel KPG-RL model that leverages keypoints to guide the correct matching in OT. We devise a mask-based constraint to preserve the matching of keypoints pairs, and propose to preserve the relation of each point to the keypoints to impose the guidance of these keypoints in OT. The effectiveness of the proposed KPG-RL model is verified in the application of HDA. The keypoint-guided OT model could be possibly applied to more applications, *e.g.*, cross-domain data generation/translation [58, 59], and point-set or image registration.

**Acknowledgements.** This work was supported by National Key R&D Program 2021YFA1003002 and NSFC (12125104, U1811461, U20B2075, 11971373, 12026605).

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
