# Keypoint-Guided Optimal Transport with Applications in Heterogeneous Domain Adaptation

## A    Mathematical Deductions

### A.1    Proof of Proposition 1

Proposition 1 in the paper is for the case that $p_i = q_j$, for all $(i, j) \in \mathcal{K}$. As stated in the paper, the mask-based modeling of the transport plan is applicable even for the case that there exist some $(i, j) \in \mathcal{K}$ such that $p_i \neq q_j$. To see this, we first mathematically give the definition of preserving the matching of keypoint pairs and then prove Proposition A-1, a generalization of Proposition 1.

**Definition A-1.** *Given the marginal distributions $\boldsymbol{p}$ and $\boldsymbol{q}$, we say that the transport plan $\pi \in \Pi(\boldsymbol{p}, \boldsymbol{q})$ preserves the matching of a keypoint pair with index $(i, j) \in \mathcal{K}$, if $\pi$ satisfies one of the following conditions:*

1. *If $p_i = q_j$, $\pi$ satisfies that $\pi_{i,j'} = 0, \forall j' \neq j; \pi_{i',j} = 0, \forall i' \neq i; \pi_{i,j} = p_i = q_j$.*

2. *If $p_i > q_j$, $\pi$ satisfies that $\pi_{i',j} = 0, \forall i' \neq i; \pi_{i,j} = q_j$.*

3. *If $p_i < q_j$, $\pi$ satisfies that $\pi_{i,j'} = 0, \forall j' \neq j; \pi_{i,j} = p_i$.*

$\square$

The left part of Fig. A-1 illustrates these conditions. Specifically, the first condition implies that if $p_i = q_j$ (*e.g.*, $(i, j)$ is taken as (4, 4) in Fig. A-1), the all mass $p_i$ of $x_i$ will be transported to $y_j$ and $y_j$ can only receive mass from $x_i$. The second condition implies that if $p_i > q_j$ (*e.g.*, $(i, j)$ is taken as (3, 2) in Fig. A-1), $y_j$ can only receive mass from $x_i$ and consequently the partial mass $p_i - q_j$ of $x_i$ is allowed to be transported to the target points apart from $y_j$. The third condition indicates that if $p_i < q_j$ (*e.g.*, $(i, j)$ is taken as (6, 5) in Fig. A-1), the all mass $p_i$ of $x_i$ will be transported to $y_j$ and $y_j$ is enabled to receive partial mass $q_j - p_i$ from the source points apart from $x_i$. For the convenience of description, for each pair $(i, j) \in \mathcal{K}$, we denote $j = \kappa(i)$ and $i = \kappa'(j)$.

**Proposition A-1.** *Suppose that the mask matrix $M$ satisfies that*

$$
M_{i,j} = \begin{cases}
1, & \text{if } (i, j) \in \mathcal{K}, \\
0, & \text{if } i \in \mathcal{I}, \ p_i \leq q_{\kappa(i)}, \ \text{and } (i, j) \notin \mathcal{K}, \\
0, & \text{if } j \in \mathcal{J}, \ p_{\kappa'(j)} \geq q_j, \ \text{and } (i, j) \notin \mathcal{K}, \\
1, & \text{if } i \in \mathcal{I}, \ p_i > q_{\kappa(i)}, \ \text{and } (i, j) \notin \mathcal{K}, \\
1, & \text{if } j \in \mathcal{J}, \ p_{\kappa'(j)} < q_j, \ \text{and } (i, j) \notin \mathcal{K}, \\
1, & \text{otherwise (i.e., } i \notin \mathcal{I}, j \notin \mathcal{J}).
\end{cases}
\tag{A-1}
$$

*Then, the transport plan $\tilde{\pi} = M \odot \pi$ with $\pi \in \Pi(\boldsymbol{p}, \boldsymbol{q}; M)$ preserves the matching of keypoint pairs with index in $\mathcal{K}$.*

According to the definition of $M$, $M_{i,j} = 1$ for the keypoint pair $(i, j) \in \mathcal{K}$, implying that $\tilde{\pi}_{i,j}$ could take non-zero value. For $i \in \mathcal{I}$, $(i, j) \notin \mathcal{K}$ and $p_i \leq q_{\kappa(i)}$, $M_{i,j}$ is set to 0, enforcing that the $i$-th row of $\tilde{\pi}$ are zeros except for the location $\kappa(i)$ of the target keypoint paired with $i$ (*e.g.*, the 4-th and 6-th rows of $\tilde{\pi}$ in Fig. A-1). Similarly, for $j \in \mathcal{J}$, $(i, j) \notin \mathcal{K}$, and $p_{\kappa'(j)} \leq q_j$, we set $M_{i,j} = 0$, enforcing that the $j$-th column of $\tilde{\pi}$ are zeros except for the location $\kappa'(j)$ of the source keypoint paired with $j$ (*e.g.*, the 2-th and 4-th columns of $\tilde{\pi}$ in Fig. A-1). For the other points (corresponding to the last three cases in Eq. (A-1)), we set $M_{i,j} = 1$, indicating that there is no additional constraint on $\tilde{\pi}_{i,j}$. If $p_i = q_j$, for all $(i, j) \in \mathcal{K}$ (*i.e.*, $p_i = q_{\kappa(i)}, \forall i \in \mathcal{I}$), Proposition A-1 degenerates to Proposition 1 in the paper.

**Proof**:
For any $(i, j) \in \mathcal{K}$, we next prove that $\tilde{\pi}$ preserves the matching of keypoint pair $(i, j)$.

- If $p_i = q_j$, from the definition of $M$, we have $M_{i,j'} = 0$ for all $j' \neq j$ and $M_{i,j} = 1$. Then, we have $\tilde{\pi}_{i,j'} = M_{i,j'}\pi_{i,j'} = 0$ for all $j' \neq j$. Since $\sum_{j'=1}^{n} \tilde{\pi}_{i,j'} = p_i$, we have $\tilde{\pi}_{i,j} = p_i$. Similarly, we have $M_{i',j} = 0$ for all $i' \neq i$. Then, we have $\tilde{\pi}_{i',j} = M_{i',j}\pi_{i',j} = 0$ for all $i' \neq i$, and $\tilde{\pi}_{i,j} = \sum_{i'=1}^{m} \tilde{\pi}_{i',j} = q_j$.

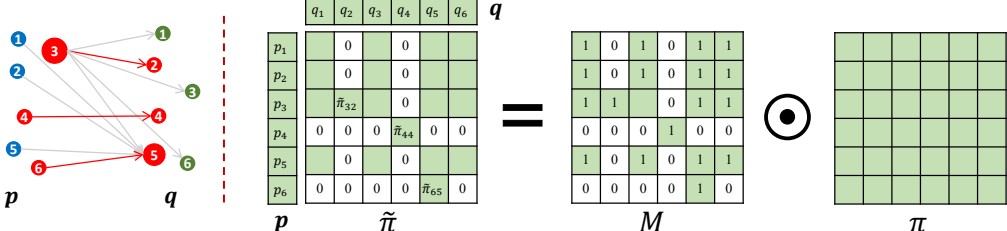

Figure A-1: Example of modeling the matching of keypoints (red) using mask, where $\mathcal{K} = \{(3,2),(4,4),(6,5)\}$ with $p_3 > q_2, p_4 = q_4, p_6 < q_5$.

- If $p_i > q_j$, from the definition of $M$, we have $M_{i',j} = 0$ for all $i' \neq i$ and $M_{i,j} = 1$. Then, we have $\tilde{\pi}_{i',j} = M_{i',j}\pi_{i',j} = 0$ for all $i' \neq i$, and $\tilde{\pi}_{i,j} = \sum_{i'=1}^{m} \tilde{\pi}_{i',j} = q_j$.
- $p_i < q_j$, from the definition of $M$, we have $M_{i,j'} = 0$ for all $j' \neq j$ and $M_{i,j} = 1$. Then $\tilde{\pi}_{i,j'} = M_{i,j'}\pi_{i,j'} = 0$ for all $j' \neq j$, and $\tilde{\pi}_{i,j} = \sum_{j'=1}^{n} \tilde{\pi}_{i,j'} = q_i$.

Thus, for any keypoint pair with index $(i,j) \in \mathcal{K}$, $\tilde{\pi}$ satisfies the conditions in Definition A-1. This means that $\tilde{\pi}$ preserves the matching of keypoint pairs with index in $\mathcal{K}$.

## A.2 Linear Programming for Solving KPG-RL

We cast the matrix $G$ (resp. $M, \pi$) as the vector $\boldsymbol{c}$ (resp. $\boldsymbol{m}, \boldsymbol{x}) \in \mathbb{R}^{mn}$, such that the $(i+m(j-1))$-th element of $\boldsymbol{c}$ is $G_{ij}$. By denoting

$$A = \begin{bmatrix} \mathbb{1}_n^\top \otimes I_m \\ I_n \otimes \mathbb{1}_m \end{bmatrix} * \operatorname{diag}(\boldsymbol{m}), \boldsymbol{h} = \begin{bmatrix} \boldsymbol{p} \\ \boldsymbol{q} \end{bmatrix}, \text{ and } \tilde{\boldsymbol{c}} = \boldsymbol{c} \odot \boldsymbol{m}, \tag{A-2}$$

where $I_m$ is the identity matrix of size $n$ and $\otimes$ is the Kronecker product, the KPG-RL model in Eq. (9) in the paper reads

$$\begin{aligned} \min_{\boldsymbol{x}} \ & \tilde{\boldsymbol{c}}^\top \boldsymbol{x} \\ s.t. \ & \boldsymbol{x} \geq 0, \\ & A\boldsymbol{x} = \boldsymbol{h}. \end{aligned} \tag{A-3}$$

With the standard form of linear programming in Eq. (A-3), the Simplex algorithm can be directly used to solve the KPG-RL model.

## A.3 Sinkhorn's Algorithm for Solving KPG-RL

The entropy-regularized model for KPG-RL is

$$\begin{aligned} \min_{\pi} \ & \langle M \odot \pi, G \rangle_F - \epsilon H(M \odot \pi) \\ s.t. \ & \pi \geq 0, (M \odot \pi)\mathbb{1}_n = \boldsymbol{p}, (M \odot \pi)^\top \mathbb{1}_m = \boldsymbol{q}, \end{aligned} \tag{A-4}$$

where $H(M \odot \pi) = -(\langle M \odot \pi, \log(M \odot \pi) \rangle_F - \mathbb{1}_m^\top (M \odot \pi)\mathbb{1}_n)$ is the entropy of the transport plan $M \odot \pi$. The Lagrangian function is

$$\begin{aligned} L(\pi, \boldsymbol{f}, \boldsymbol{g}) = & \langle M \odot \pi, G \rangle_F + \epsilon \left( \langle M \odot \pi, \log(M \odot \pi) \rangle_F - \mathbb{1}_m^\top (M \odot \pi)\mathbb{1}_n \right) \\ & - \langle \boldsymbol{f}, (M \odot \pi)\mathbb{1}_n - \boldsymbol{p} \rangle_F - \langle \boldsymbol{g}, (M \odot \pi)^\top \mathbb{1}_m - \boldsymbol{q} \rangle_F, \end{aligned} \tag{A-5}$$

where $\boldsymbol{f} \in \mathbb{R}^m$ and $\boldsymbol{g} \in \mathbb{R}^n$. The first-order conditions then yield

$$\frac{\partial L}{\partial \pi_{i,j}} = M_{i,j}G_{i,j} + \epsilon M_{i,j}\log(M_{i,j}\pi_{i,j}) - M_{i,j}f_i - M_{i,j}g_j = 0. \tag{A-6}$$

If $M_{i,j} = 0$, $\pi_{i,j}$ could be arbitrary non-negative value, and if $M_{i,j} = 1$, we have $\pi_{i,j} = e^{f_i/\epsilon}e^{-G_{i,j}/\epsilon}e^{g_j/\epsilon}$. Therefore, we can unify the expression as $\pi_{ij} = M_{ij}e^{f_i/\epsilon}e^{-C_{ij}/\epsilon}e^{g_j/\epsilon}$, in

which we enforce $\pi_{i,j} = 0$ if $M_{i,j} = 0$. The matrix form is $\pi = \mathrm{diag}(\boldsymbol{u})K\mathrm{diag}(\boldsymbol{v})$ where $\boldsymbol{u} = e^{\boldsymbol{f}/\epsilon}, K = M \odot e^{-G/\epsilon}$, and $\boldsymbol{v} = e^{\boldsymbol{g}/\epsilon}$. The constraints are

$$\mathrm{diag}(\boldsymbol{u})K\mathrm{diag}(\boldsymbol{v})\mathbb{1}_n = \boldsymbol{p}, \quad (\mathrm{diag}(\boldsymbol{u})K\mathrm{diag}(\boldsymbol{v}))^\top \mathbb{1}_m = \boldsymbol{q}. \tag{A-7}$$

Since the entries of $K$ are non-negative, the Sinkhorn's algorithm can be applied [15]. The iteration formulas are

$$\boldsymbol{u}^{(l+1)} = \frac{\boldsymbol{p}}{K\boldsymbol{v}^l}, \quad \boldsymbol{v}^{(l+1)} = \frac{\boldsymbol{q}}{K^\top \boldsymbol{u}^{(l+1)}}. \tag{A-8}$$

The division operator used above is entry-wise.

**Log-domain Sinkhorn iteration.** For KP, the Sinkhorn iteration in the log-domain is more stable [60]. We next deduce the log-domain Sinkhorn iteration for our KPG-RL. In the log-domain, the left equation in Eq. (A-8) is

$$\frac{1}{\epsilon} f_i^{(l+1)} = \log(p_i) - \log\left(\sum_{j=1}^n M_{i,j} e^{\frac{-G_{i,j}+g_j^{(l)}}{\epsilon}}\right). \tag{A-9}$$

Let $H(\boldsymbol{f},\boldsymbol{g}) = \frac{1}{\epsilon}(-G + \boldsymbol{f}\mathbb{1}_n^\top + \boldsymbol{g}\mathbb{1}_m^\top)$, then

$$\begin{aligned}
f_i^{(l+1)} &= \epsilon\log(p_i) - \epsilon\log\left(\sum_{j=1}^n M_{i,j} e^{H(\boldsymbol{f}^{(l)},\boldsymbol{g}^{(l)})_{i,j}} e^{-f_i^{(l)}/\epsilon}\right) \\
&= \epsilon\log(p_i) - \epsilon\log\left(e^{-f_i^{(l)}/\epsilon}\sum_{j=1}^n M_{i,j} e^{H(\boldsymbol{f}^{(l)},\boldsymbol{g}^{(l)})_{i,j}}\right) \\
&= \epsilon\log(p_i) - \epsilon\log\left(\sum_{j=1}^n M_{i,j} e^{H(\boldsymbol{f}^{(l)},\boldsymbol{g}^{(l)})_{i,j}}\right) + f_i^{(l)} \\
&= \epsilon\log(p_i) - \epsilon\log\left(\sum_{j=1}^n e^{\log(M_{i,j})H(\boldsymbol{f}^{(l)},\boldsymbol{g}^{(l)})_{i,j}}\right) + f_i^{(l)}
\end{aligned} \tag{A-10}$$

If $M_{i,j} = 0$, $\log(M_{i,j})H(\boldsymbol{f}^{(l)},\boldsymbol{g}^{(l)})_{i,j} = -\infty$. We define $\bar{H}(\boldsymbol{f},\boldsymbol{g})$ as

$$\bar{H}(\boldsymbol{f},\boldsymbol{g})_{i,j} = \begin{cases} H(\boldsymbol{f},\boldsymbol{g})_{i,j} & \text{if } M_{i,j} = 1, \\ -\infty & \text{if } M_{i,j} = 0, \end{cases} \tag{A-11}$$

and define the log-sum-exp function $\mathrm{LogSumExp}: \mathbb{R}^{m\times n} \to \mathbb{R}^m$ as

$$\mathrm{LogSumExp}(A) = \left(\log(\sum_j e^{A_{1,j}}), \log(\sum_j e^{A_{2,j}}), \cdots, \log(\sum_j e^{A_{m,j}})\right)^\top. \tag{A-12}$$

Then, the matrix form of Eq. (A-10) becomes

$$\boldsymbol{f}^{(l+1)} = \epsilon\log(\boldsymbol{p}) - \epsilon\mathrm{LogSumExp}\left(\bar{H}(\boldsymbol{f}^{(l)},\boldsymbol{g}^{(l)})\right) + \boldsymbol{f}^{(l)}. \tag{A-13}$$

Similarly, the corresponding iteration formula in the log-domain of the right equation in (A-8) is

$$\boldsymbol{g}^{(l+1)} = \epsilon\log(\boldsymbol{q}) - \epsilon\mathrm{LogSumExp}\left(\bar{H}(\boldsymbol{f}^{(l+1)},\boldsymbol{g}^{(l)})^\top\right) + \boldsymbol{g}^{(l)}. \tag{A-14}$$

Equations (A-13) and (A-14) consists in the formulas of the Sinkhorn iteration in the log-domain.

## A.4 Frank-Walfe Algorithm for Solving KPG-RL-GW

We define the 4-order tensor $L = (L_{i,j,k,l}) \in \mathbb{R}^{m\times n\times m\times n}$ by $L_{i,j,k,l} = (C_{i,k}^s - C_{j,l}^t)^2$, and define the tensor-matrix product $L \circ \pi = ((L \circ \pi)_{i,j}) \in \mathbb{R}^{m\times n}$ by $(L \circ \pi)_{i,j} = \sum_{k,l} L_{i,j,k,l}\pi_{k,l}$. Then, The KPG-RL-GW model in Eq. (12) in the paper reads

$$\min_{\pi\in\Pi(\boldsymbol{p},\boldsymbol{q};M)} \langle(M \odot \pi), \alpha L \circ (M \odot \pi) + (1-\alpha)G\rangle_F \triangleq \mathcal{L}(\pi) \tag{A-15}$$

The gradient of the objective function is

$$\nabla \mathcal{L}(\pi) = M \odot (2\alpha L \circ (M \odot \pi) + (1 - \alpha)G). \tag{A-16}$$

In $k$-th iteration of the Frank-Walfe algorithm, it runs the following three steps:

**Step 1.** Compute a linear minimization oracle over the set $\Pi(\boldsymbol{p}, \boldsymbol{q}; M)$, *i.e.*,

$$\hat{\pi} \leftarrow \underset{\pi \in \Pi(\boldsymbol{p}, \boldsymbol{q}; M)}{\operatorname{argmin}} \langle \nabla \mathcal{L}(\pi^{(k)}), \pi \rangle_F. \tag{A-17}$$

Equation (A-17) can be rewritten as

$$\hat{\pi} \leftarrow \underset{\pi \in \Pi(\boldsymbol{p}, \boldsymbol{q}; M)}{\operatorname{argmin}} \langle M \odot \pi, 2\alpha L \circ (M \odot \pi^{(k)}) + (1 - \alpha)G \rangle_F, \tag{A-18}$$

Equation (A-18) is a KPG-RL-like problem and can be solved using linear programming or Sinkhorn's algorithm.

**Step 2.** Determine optimal step-size $\beta^{(k)}$ subject to

$$\beta^{(k)} \leftarrow \underset{\beta \in [0,1]}{\operatorname{argmin}} \mathcal{L}((1 - \beta)\pi^{(k)} + \beta\hat{\pi}). \tag{A-19}$$

$\beta^{(k)}$ can be obtained by the line-search method in [61].

**Step 3.** Update

$$\pi^{(k+1)} = (1 - \beta^{(k)})\pi^{(k)} + \beta^{(k)}\hat{\pi}. \tag{A-20}$$

## A.5 Theoretical Properties of KPG-RL-KP and KPG-RL-GW

In this section, we show that given prior "correct" paired keypoints, the KPG-RL-KP model provides a proper metric for distributions supported in the same space, and the the KPG-RL-GW model provides a divergence for distributions in distinct spaces, under mild conditions. Since the discrete distributions $\boldsymbol{p} = \frac{1}{m}\sum_i^m \delta_{x_i}$ (resp. $\boldsymbol{q} = \frac{1}{n}\sum_j^n \delta_{y_j}$) are invariant to the permutation of $\{x_i\}_{i=1}^m$ (resp. $\{y_j\}_{j=1}^n$), we assume that any two paired keypoints across domains share the same index. Therefore, the index set of paired keypoints is $\mathcal{K} = \{(i_u, i_u)\}_{u=1}^U$. We assume $p_{i_u} = q_{i_u}, \forall i_u$, in this section. For the convenience of description, in this section, we denote $M^{\boldsymbol{pq}}$ as the mask matrix for transporting $\boldsymbol{p}$ to $\boldsymbol{q}$, and $\mathcal{P}_{\mathcal{I}}^{\mathcal{X}}$ as the set of discrete probability distributions supported on $m$ points in ground space $\mathcal{X}$ such that all distributions in $\mathcal{P}_{\mathcal{I}}^{\mathcal{X}}$ share the keypoint index set $\mathcal{I} = \{i_u\}_{u=1}^U$.

### A.5.1 KPG-RL-KP Providing a Proper Metric

For distributions supported in the same space, the "correct" paired keypoints indicates that if $\boldsymbol{p} = \boldsymbol{q}$, each source keypoint is equal to its paired target keypoint, *i.e.*, $x_{i_u} = y_{i_u}$, for any $i_u \in \mathcal{I}$. We denote

$$\mathcal{S}_{krk}(\boldsymbol{p}, \boldsymbol{q}) = \min_{\pi \in \Pi(\boldsymbol{p}, \boldsymbol{q}; M^{\boldsymbol{pq}})} \sum_{i,j} M_{i,j}^{\boldsymbol{pq}} \pi_{i,j}(\alpha C_{i,j} + (1 - \alpha)G_{i,j}), \tag{A-21}$$

where $\alpha \in (0, 1)$.

**Theorem A-1.** *Suppose $c$ is a proper distance in space $\mathcal{X}$ and $d$ is a proper distance in probability simplex $\Sigma_m$. Then, for any $\boldsymbol{p}$ and $\boldsymbol{q}$ in $\mathcal{P}_{\mathcal{I}}^{\mathcal{X}}$, given the "correct" paired keypoints stated above, $\mathcal{S}_{krk}(\boldsymbol{p}, \boldsymbol{q})$ is a proper distance between $\boldsymbol{p}$ and $\boldsymbol{q}$.*

**Proof**:

(1) ***Show that*** $\mathcal{S}_{krk}(\boldsymbol{p}, \boldsymbol{q}) = 0$ ***if and only if*** $\boldsymbol{p} = \boldsymbol{q}$. (a) If $\boldsymbol{p} = \boldsymbol{q}$, we have $x_i = y_i$ and $p_i = q_i$ for any $i \in [m]$ (since the permutation of support points does not change the distribution). Hence, $C_{i,i} = c(x_i, y_i) = 0$, and $C_{i,i_u}^s = c(x_i, x_{i_u}) = c(y_i, y_{i_u}) = C_{i,i_u}^t, \forall i \in [m]$ and $\forall i_u \in \mathcal{I}$, which implies that $R_i^s = R_i^t$. Then, we have $G_{i,i} = d(R_i^s, R_i^t) = 0$. We define $\pi$ by $\pi_{i,j} = p_i$ if $i = j$, and 0 otherwise. Obviously, $M^{\boldsymbol{pq}} \odot \pi$ is in $\Pi(\boldsymbol{p}, \boldsymbol{q}; M^{\boldsymbol{pq}})$ and $\sum_{i,j} M_{i,j}^{\boldsymbol{pq}} \pi_{i,j}(\alpha C_{i,j} + (1 - \alpha)G_{i,j}) = 0$. Therefore, $\mathcal{S}_{krk}(\boldsymbol{p}, \boldsymbol{q}) = 0$. (b) We denote $\pi^*$ as the optimal solution of problem (A-21). If $\mathcal{S}_{krk}(\boldsymbol{p}, \boldsymbol{q}) = 0$, we have $\langle M^{\boldsymbol{pq}} \odot \pi^*, C \rangle_F = 0$. This means that the KP problem $\min_{\pi \in \Pi(\boldsymbol{p}, \boldsymbol{q})} \langle \pi, C \rangle_F = 0$. Using the Proposition 2.2 in [62], we have $\boldsymbol{p} = \boldsymbol{q}$.

(2) **Show that** $\mathcal{S}_{krk}(\boldsymbol{p}, \boldsymbol{q}) = \mathcal{S}_{krk}(\boldsymbol{q}, \boldsymbol{p})$. From the definition of mask matrix in Proposition 1 in the paper, we have $M_{i,j}^{\boldsymbol{pq}} = M_{j,i}^{\boldsymbol{qp}}$. $C$ and $G$ are symmetric because $c$ and $d$ are distances. For any $\pi \in \Pi(\boldsymbol{p}, \boldsymbol{q}; M^{\boldsymbol{pq}})$, we define $\pi'$ as $\pi'_{i,j} = \pi_{j,i}$, and then $\pi' \in \Pi(\boldsymbol{q}, \boldsymbol{p}; M^{\boldsymbol{qp}})$. Then, we have

$$
\begin{aligned}
\mathcal{S}_{krk}(\boldsymbol{p}, \boldsymbol{q}) &= \min_{\pi \in \Pi(\boldsymbol{p}, \boldsymbol{q}; M^{\boldsymbol{pq}})} \sum_{i,j} M_{i,j}^{\boldsymbol{pq}} \pi_{i,j} (\alpha C_{i,j} + (1-\alpha) G_{i,j}) \\
&= \min_{\pi \in \Pi(\boldsymbol{p}, \boldsymbol{q}; M^{\boldsymbol{pq}})} \sum_{i,j} M_{j,i}^{\boldsymbol{qp}} \pi_{i,j} (\alpha C_{j,i} + (1-\alpha) G_{j,i}) \\
&= \min_{\pi' \in \Pi(\boldsymbol{q}, \boldsymbol{p}; M^{\boldsymbol{qp}})} \sum_{j,i} M_{j,i}^{\boldsymbol{qp}} \pi'_{j,i} (\alpha C_{j,i} + (1-\alpha) G_{j,i}) \\
&= \mathcal{S}_{krk}(\boldsymbol{q}, \boldsymbol{p}).
\end{aligned} \tag{A-22}
$$

(3) **Show that** $\mathcal{S}_{krk}(\boldsymbol{p}, \boldsymbol{q}) \leq \mathcal{S}_{krk}(\boldsymbol{p}, \boldsymbol{r}) + \mathcal{S}_{krk}(\boldsymbol{r}, \boldsymbol{q})$. Let $M^{\boldsymbol{pr}} \odot \pi^{\boldsymbol{pr}}$ and $M^{\boldsymbol{rq}} \odot \pi^{\boldsymbol{rq}}$ be the optimal transport plans corresponding to $\mathcal{S}_{krk}(\boldsymbol{p}, \boldsymbol{r})$ and $\mathcal{S}_{krk}(\boldsymbol{r}, \boldsymbol{q})$, respectively. We define

$$
\tilde{\gamma} = (M^{\boldsymbol{pr}} \odot \pi^{\boldsymbol{pr}}) \operatorname{diag}\left(\frac{1}{\tilde{\boldsymbol{r}}}\right) (M^{\boldsymbol{rq}} \odot \pi^{\boldsymbol{rq}}), \tag{A-23}
$$

where the element $\tilde{r}_j$ of $\tilde{\boldsymbol{r}}$ is $r_j$ if $r_j > 0$, and 1 otherwise. We notice that

$$
\tilde{\gamma} \mathbb{1}_m = (M^{\boldsymbol{pr}} \odot \pi^{\boldsymbol{pr}}) \operatorname{diag}\left(\frac{1}{\tilde{\boldsymbol{r}}}\right) \boldsymbol{r} = (M^{\boldsymbol{pr}} \odot \pi^{\boldsymbol{pr}}) \left(\frac{\boldsymbol{r}}{\tilde{\boldsymbol{r}}}\right) = (M^{\boldsymbol{pr}} \odot \pi^{\boldsymbol{pr}}) \tilde{\mathbb{1}}_m, \tag{A-24}
$$

where the $j$-th location of $\tilde{\mathbb{1}}_m$ is 1 if $r_j > 0$, and 0 otherwise. Note that for $j$ such that $r_j = 0$, we have $\sum_{i,j} (M^{\boldsymbol{pr}} \odot \pi^{\boldsymbol{pr}})_{i,j} = r_j = 0$, which implies $(M^{\boldsymbol{pr}} \odot \pi^{\boldsymbol{pr}})_{i,j} = 0$ for any $i$. Hence,

$$
(M^{\boldsymbol{pr}} \odot \pi^{\boldsymbol{pr}}) \tilde{\mathbb{1}}_m = (M^{\boldsymbol{pr}} \odot \pi^{\boldsymbol{pr}}) \mathbb{1}_m = \boldsymbol{p}. \tag{A-25}
$$

Similarity, $\tilde{\gamma}^\top \mathbb{1}_m = \boldsymbol{q}$. Since the indexes of paired keypoints across any two distribution in $\mathcal{P}_{\mathcal{I}}^{\mathcal{X}}$ are the same, for any $i_u \in \mathcal{I}$, the $i_u$-th row and column of $M^{\boldsymbol{pr}}$ and $M^{\boldsymbol{rq}}$ are zeros except for that $M_{i_u, i_u}^{\boldsymbol{pr}} = M_{i_u, i_u}^{\boldsymbol{rq}} = 1$. So the $i_u$-th row and column of $\tilde{\gamma}$ are zeros except for $\tilde{\gamma}_{i_u, i_u}$. Then, we can write $\tilde{\gamma}_{i_u, i_u} = M^{\boldsymbol{pq}} \odot \gamma$ with $\gamma \in \mathbb{R}_+^{m \times m}$. Further, we have $\gamma \in \Pi(\boldsymbol{p}, \boldsymbol{q}; M^{\boldsymbol{pq}})$. The triangle inequality follows then from

$$
\mathcal{S}_{krk}(\boldsymbol{p}, \boldsymbol{q}) = \min_{\pi \in \Pi(\boldsymbol{p}, \boldsymbol{q}; M^{\boldsymbol{pq}})} \sum_{i,j} M_{i,j}^{\boldsymbol{pq}} \pi_{i,j} (\alpha c(x_i, y_j) + (1-\alpha) d(R_i^s, R_j^t))
$$

$$
\leq \sum_{i,j} \tilde{\gamma}_{i,j} (\alpha c(x_i, y_j) + (1-\alpha) d(R_i^s, R_j^t))
$$

$$
= \sum_{i,j} (\alpha c(x_i, y_j) + (1-\alpha) d(R_i^s, R_j^t)) \sum_k \frac{(M^{\boldsymbol{pr}} \odot \pi^{\boldsymbol{pr}})_{i,k} (M^{\boldsymbol{rq}} \odot \pi^{\boldsymbol{rq}})_{k,j}}{\tilde{r}_k}
$$

$$
\leq \sum_{i,k,j} (\alpha(c(x_i, z_k) + c(z_k, y_j)) + (1-\alpha)(d(R_i^s, R_k^r) + d(R_k^r, R_j^t))) \frac{(M^{\boldsymbol{pr}} \odot \pi^{\boldsymbol{pr}})_{i,k} (M^{\boldsymbol{rq}} \odot \pi^{\boldsymbol{rq}})_{k,j}}{\tilde{r}_k}
$$

$$
= \sum_{i,k,j} (\alpha c(x_i, z_k) + (1-\alpha) d(R_i^s, R_k^r)) \frac{(M^{\boldsymbol{pr}} \odot \pi^{\boldsymbol{pr}})_{i,k} (M^{\boldsymbol{rq}} \odot \pi^{\boldsymbol{rq}})_{k,j}}{\tilde{r}_k}
$$

$$
+ \sum_{i,k,j} (\alpha c(z_k, y_j) + (1-\alpha) d(R_k^r, R_j^t)) \frac{(M^{\boldsymbol{pr}} \odot \pi^{\boldsymbol{pr}})_{i,k} (M^{\boldsymbol{rq}} \odot \pi^{\boldsymbol{rq}})_{k,j}}{\tilde{r}_k}
$$

$$
= \sum_{i,k} (\alpha c(x_i, z_k) + (1-\alpha) d(R_i^s, R_k^r)) (M^{\boldsymbol{pr}} \odot \pi^{\boldsymbol{pr}})_{i,k}
$$

$$
+ \sum_{k,j} (\alpha c(z_k, y_j) + (1-\alpha) d(R_k^r, R_j^t)) (M^{\boldsymbol{rq}} \odot \pi^{\boldsymbol{rq}})_{k,j}
$$

$$
= \mathcal{S}_{krk}(\boldsymbol{p}, \boldsymbol{r}) + \mathcal{S}_{krk}(\boldsymbol{r}, \boldsymbol{q}), \tag{A-26}
$$

where $z_k$ is the support point of $\boldsymbol{r}$ and $R_k^r$ is the relation of $z_k$ to the keypoints of $\boldsymbol{r}$.

$\square$

### A.5.2 KPG-RL-GW Providing a Divergence

For any distribution $\boldsymbol{p} \in \mathcal{P}_{\mathcal{I}}^{\mathcal{X}}$ and $\boldsymbol{q} \in \mathcal{P}_{\mathcal{I}}^{\mathcal{Y}}$, $\boldsymbol{p}$ and $\boldsymbol{q}$ are said to be isomorphic if there exists a bijection $\sigma : [m] \longmapsto [m]$ such that $c(x_i, x_k) = c'(y_{\sigma(i)}, y_{\sigma(k)})$, and $p_i = q_{\sigma(i)}$, where $[m] = \{1, 2, \cdots, m\}$, and $c$ and $c'$ are respectively proper distances in $\mathcal{X}$ and $\mathcal{Y}$. The keypoints are "correct" means that if $\boldsymbol{p} = \boldsymbol{q}$, $\sigma$ maps each source keypoint to its paired target keypoint, *i.e.*, $\sigma(i_u) = i_u$, for any $i_u \in \mathcal{I}$. We denote

$$
\mathcal{S}_{krg}(\boldsymbol{p}, \boldsymbol{q}) = \min_{\pi \in \Pi(\boldsymbol{p}, \boldsymbol{q}; M^{\boldsymbol{pq}})} \sum_{i,j} \Big[ \alpha \Big( \sum_{k,l} (M^{\boldsymbol{pq}} \odot \pi)_{i,j} (M^{\boldsymbol{pq}} \odot \pi)_{k,l} |C_{i,k}^s - C_{j,l}^t|^2 \Big) \\
+ (1 - \alpha)(M^{\boldsymbol{pq}} \odot \pi)_{i,j} G_{i,j} \Big]. \tag{A-27}
$$

**Theorem A-2.** *Suppose $c$ and $c'$ are proper distances in spaces $\mathcal{X}$ and $\mathcal{Y}$. Suppose $d$ is a divergence in probability simplex $\Sigma_m$. Then, for any $\boldsymbol{p}$ in $\mathcal{P}_{\mathcal{I}}^{\mathcal{X}}$ and any $\boldsymbol{q}$ in $\mathcal{P}_{\mathcal{I}}^{\mathcal{Y}}$, given the "correct" paired keypoints stated above, $\mathcal{S}_{krg}(\boldsymbol{p}, \boldsymbol{q}) = 0$ if and only if $\boldsymbol{p}$ and $\boldsymbol{q}$ are isomorphic.*

**Proof:**

(a) If $\boldsymbol{p}$ and $\boldsymbol{q}$ are isomorphic, for any $i \in [m]$ and any $i_u \in \mathcal{I}$, we have $c(x_i, x_{i_u}) = c'(y_{\sigma(i)}, y_{\sigma(i_u)}) = c'(y_{\sigma(i)}, y_{i_u})$, implying that $R_i^s = R_{\sigma(i)}^t$. We define $\pi$ as $\pi_{i,j} = p_i$ if $j = \sigma(i)$, and 0 otherwise. We then have

$$
\sum_{i,j} \alpha \Big( \sum_{k,l} (M^{\boldsymbol{pq}} \odot \pi)_{i,j} (M^{\boldsymbol{pq}} \odot \pi)_{k,l} |C_{i,k}^s - C_{j,l}^t|^2 \Big) + (1 - \alpha)(M^{\boldsymbol{pq}} \odot \pi)_{i,j} G_{i,j}
$$
$$
= \sum_i \Big[ \alpha \Big( \sum_k (M^{\boldsymbol{pq}} \odot \pi)_{i,\sigma(i)} (M^{\boldsymbol{pq}} \odot \pi)_{k,\sigma(k)} |C_{i,k}^s - C_{\sigma(i),\sigma(k)}^t|^2 \Big)
$$
$$
+ (1 - \alpha)(M^{\boldsymbol{pq}} \odot \pi)_{i,\sigma(i)} G_{i,\sigma(i)} \Big] \tag{A-28}
$$
$$
= \sum_i \Big[ \alpha \Big( \sum_k (M^{\boldsymbol{pq}} \odot \pi)_{i,\sigma(i)} (M^{\boldsymbol{pq}} \odot \pi)_{k,\sigma(k)} |c(x_i, x_k) - c'(x_{\sigma(i)}, x_{\sigma(k)})|^2 \Big)
$$
$$
+ (1 - \alpha)(M^{\boldsymbol{pq}} \odot \pi)_{i,\sigma(i)} d(R_i^s, R_{\sigma(i)}^t) \Big]
$$
$$
= 0.
$$

This implies $\mathcal{S}_{krg}(\boldsymbol{p}, \boldsymbol{q}) = 0$.

(b) Let $(M^{\boldsymbol{pq}}) \odot \pi^*$ be the optimal transport plan corresponding to $\mathcal{S}_{krg}(\boldsymbol{p}, \boldsymbol{q}) = 0$. If $\mathcal{S}_{krg}(\boldsymbol{p}, \boldsymbol{q}) = 0$, we have

$$
\sum_{i,j,k,l} (M^{\boldsymbol{pq}} \odot \pi^*)_{i,j} (M^{\boldsymbol{pq}} \odot \pi^*)_{k,l} |C_{i,k}^s - C_{j,l}^t|^2 = 0. \tag{A-29}
$$

This indicates that the Gromov-Wasserstein distance

$$
\min_{\pi \in \Pi(\boldsymbol{p}, \boldsymbol{q})} \sum_{i,j,k,l} \pi_{i,j} \pi_{k,l} |C_{i,k}^s - C_{j,l}^t|^2 = 0. \tag{A-30}
$$

By virtue to Gromov-Wasserstein properties in [27], there exists a bijection $\sigma : [m] \longmapsto [m]$ such that $c(x_i, x_k) = c'(y_{\sigma(i)}, y_{\sigma(k)})$, and $p_i = q_{\sigma(i)}$. $\qquad \square$

From Theorem A-2, the KPG-RL-GW model provides a divergence in the sense of isomorphism.

### A.6 Motivations of the Solving Algorithm for Partial-KPG-RL Model

In the partial-KPG-RL model in Eq. (13) in the paper, only $s$-unit mass of source and target distributions is matched. Inspired by [20], we add a dummy point with mass $\|\boldsymbol{q}\|_1 - s$ for source domain (the left black circle in Fig. A-2) and a dummy point with mass $\|\boldsymbol{p}\|_1 - s$ for target domain (the right black circle in Fig. A-2). We denote $\bar{\boldsymbol{p}} = (\boldsymbol{p}^\top, \|\boldsymbol{q}\|_1 - s)^\top$ and $\bar{\boldsymbol{q}} = (\boldsymbol{q}^\top, \|\boldsymbol{p}\|_1 - s)^\top$. As illustrated in Fig. A-2, we aim to design the extended guiding matrix $\bar{G}$ and extended mask matrix $\bar{M}$ such that performing KPG-RL between $\bar{\boldsymbol{p}}$ and $\bar{\boldsymbol{q}}$ will transport $\|\boldsymbol{p}\|_1 - s$ mass from source real data points to the target dummy point and transport $\|\boldsymbol{q}\|_1 - s$ mass from source dummy point to target real data

points. As a sequence, only $s$ mass of source and target real data points are matched. Meanwhile, the keypoints should not be matched to the dummy points because they are annotated data to guide the matching. To do this, we extend $G, M$ by

$$\bar{G} = \begin{bmatrix} G & \xi\mathbb{1}_n \\ \xi\mathbb{1}_m^\top & 2\xi + A \end{bmatrix}, \bar{M} = \begin{bmatrix} M & \boldsymbol{a} \\ \boldsymbol{b}^\top & 1 \end{bmatrix},$$

where $A > 0, \xi > 0, \boldsymbol{a} \in \mathbb{R}^m, \boldsymbol{b} \in \mathbb{R}^n$, and $M$ is constructed as in Proposition A-1. The element $a_i$ of $\boldsymbol{a}$ is 0 if $i \in \mathcal{I}$, and 1 otherwise. The element $b_j$ of $\boldsymbol{b}$ is 0 if $j \in \mathcal{J}$, and 1 otherwise. By Theorem A-3, solving partial-KPG-RL model boils down to solving the KPG-RL-like problem $\min_{\bar{\pi} \in \Pi(\bar{\boldsymbol{p}}, \bar{\boldsymbol{q}}; \bar{M})} \langle \bar{M} \odot \bar{\pi}, \bar{G} \rangle_F$.

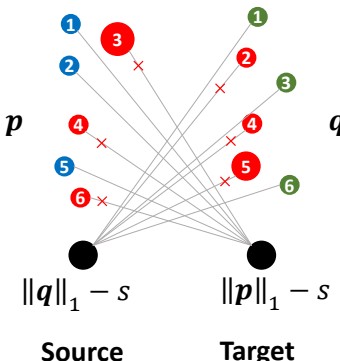

Figure A-2: Illustration of dummy points (black circles) for source and target domains.

## A.7 Proof of Theorem 1

For the convenience of understanding, the Theorem 1 in the paper is for the case that $p_i = q_j$ for all $(i, j) \in \mathcal{K}$. In this appendix, we provide and prove the Theorem A-3, a generalization of Theorem 1 in the paper, for the general case that there could exist some $(i, j) \in \mathcal{K}$ such that $p_i \neq q_j$. Before that, we rewrite the partial-KPG-RL model first.

**Partial-KPG-RL model:**

$$\min_{\pi \in \Pi^s(\boldsymbol{p}, \boldsymbol{q}; M)} \left\{ L_{kpg}(M \odot \pi) = \langle M \odot \pi, G \rangle_F \right\}, \tag{A-31}$$

where $\Pi^s(\boldsymbol{p}, \boldsymbol{q}; M) = \{\pi \in \mathbb{R}_+^{m \times n} | (M \odot \pi)\mathbb{1}_n \leqslant \boldsymbol{p}, (M \odot \pi)^\top \mathbb{1}_m \leqslant \boldsymbol{q}, \mathbb{1}_m^\top (M \odot \pi)\mathbb{1}_n = s; (M \odot \pi)_{i,:}\mathbb{1}_n = p_i, \forall i \in \mathcal{I}; \mathbb{1}_m^\top (M \odot \pi)_{:,j} = q_j, \forall j \in \mathcal{J}\}$.

**Theorem A-3.** *Suppose $A > 0$, $\xi > 0$, $\sum_{i \in \mathcal{I}} p_i + \max\{q_{\kappa(i)} - p_i, 0\} < s$, and $\sum_{j \in \mathcal{J}} q_j + \max\{p_{\kappa'(j)} - q_j, 0\} < s$, then the optimal transport plan $M \odot \pi^*$ of partial-KPG-RL model is the $m$-by-$n$ block in the upper left corner of the optimal transport plan $\bar{M} \odot \bar{\pi}^*$ of problem $\min_{\bar{\pi} \in \Pi(\bar{\boldsymbol{p}}, \bar{\boldsymbol{q}}; \bar{M})} \langle \bar{M} \odot \bar{\pi}, \bar{G} \rangle_F$.*

The definitions of $\kappa(i)$ and $\kappa'(j)$ are given in Appendix A.1. The condition $\sum_{i \in \mathcal{I}} p_i + \max\{q_{\kappa(i)} - p_i, 0\} < s$ implies that the sum of the mass ($\sum_{i \in \mathcal{I}} p_i$) of source keypoints and the mass ($\sum_{i \in \mathcal{I}} \max\{q_{\kappa(i)} - p_i, 0\}$) of the other source points apart from keypoints that should be transported to target keypoints is less than $s$. The condition $\sum_{j \in \mathcal{J}} q_j + \max\{p_{\kappa'(j)} - q_j, 0\} < s$ implies that the sum of the mass ($\sum_{j \in \mathcal{J}} q_j$) of target keypoints and the mass ($\sum_{j \in \mathcal{J}} \max\{p_{\kappa'(j)} - q_j, 0\}$) of the other target points apart from keypoints received from source keypoints is less than $s$. The two conditions are reasonable to guarantee the admissible solutions of problem (A-31). If $p_i = q_j$ for all $(i, j) \in \mathcal{K}$ (*i.e.*, $p_i = q_{\kappa(i)}, \forall i \in \mathcal{I}$ and $q_j = p_{\kappa'(j)}, \forall j \in \mathcal{J}$), Theorem A-3 degenerates to the Theorem 1 in the paper.

### A.7.1 Proof

We denote $\ddot{\pi} = \bar{\pi}_{1:m,1:n}^*$ and $t = \bar{\pi}_{m+1,n+1}^*$. To prove Theorem A-3, we first give some preparations, and then conduct the following three steps. In step 1, we show that $t = 0$. In step 2, we show that

$\ddot{\pi} \in \Pi^s(\boldsymbol{p}, \boldsymbol{q}; M)$ which means that $\ddot{\pi}$ is a feasible solution of problem (A-31). In step 3, we show that $M \odot \ddot{\pi}$ is the optimal transport plan of problem (A-31). We next detail these steps.

**Preparations:**

Since $\bar{\pi}^* \in \Pi(\bar{\boldsymbol{p}}, \bar{\boldsymbol{q}}; \bar{M})$ and $t = \bar{\pi}^*_{m+1,n+1}$, we have

$$
\begin{aligned}
\mathbb{1}_{m+1}^\top (\bar{M} \odot \bar{\pi}^*) \mathbb{1}_{n+1} &= \begin{bmatrix} \mathbb{1}_m^\top & 1 \end{bmatrix} \left( \begin{bmatrix} M & \boldsymbol{a} \\ \boldsymbol{b} & 1 \end{bmatrix} \odot \begin{bmatrix} \ddot{\pi} & \bar{\pi}^*_{1:m,n+1} \\ \bar{\pi}^*_{m+1,1:n} & \bar{\pi}^*_{m+1,n+1} \end{bmatrix} \right) \begin{bmatrix} \mathbb{1}_n \\ 1 \end{bmatrix} \\
&= \mathbb{1}_m^\top (M \odot \ddot{\pi}) \mathbb{1}_n + \sum_{i=1}^m a_i \bar{\pi}^*_{i,n+1} + \sum_{j=1}^n b_j \bar{\pi}^*_{m+1,j} + t.
\end{aligned}
\tag{A-32}
$$

Meanwhile, we have

$$
\mathbb{1}_{m+1}^\top (\bar{M} \odot \bar{\pi}^*) \mathbb{1}_{n+1} = \mathbb{1}_{m+1}^\top \bar{\boldsymbol{p}} = \|\bar{\boldsymbol{p}}\|_1 = \|\boldsymbol{p}\|_1 + \|\boldsymbol{q}\|_1 - s,
\tag{A-33}
$$

$$
\sum_{i=1}^m a_i \bar{\pi}^*_{i,n+1} + t = \|\boldsymbol{p}\|_1 - s,
\tag{A-34}
$$

and

$$
\sum_{j=1}^n b_j \bar{\pi}^*_{m+1,j} + t = \|\boldsymbol{q}\|_1 - s.
\tag{A-35}
$$

Combining the above four equations, we have

$$
\mathbb{1}_m^\top (M \odot \ddot{\pi}) \mathbb{1}_n + \|\boldsymbol{p}\|_1 + \|\boldsymbol{q}\|_1 - 2s - t = \|\boldsymbol{p}\|_1 + \|\boldsymbol{q}\|_1 - s.
\tag{A-36}
$$

Therefore,

$$
\mathbb{1}_m^\top (M \odot \ddot{\pi}) \mathbb{1}_n = s + t.
\tag{A-37}
$$

**Step 1: show that $t = \bar{\pi}^*_{m+1,n+1} = 0$.**

First, we have

$$
\begin{aligned}
\langle \bar{M} \odot \bar{\pi}^*, \bar{G} \rangle_F &= \sum_{i=1}^m \sum_{j=1}^n M_{i,j} \bar{\pi}^*_{i,j} G_{i,j} + \xi \sum_{i=1}^m a_i \bar{\pi}^*_{i,n+1} \\
&\quad + \xi \sum_{j=1}^n b_j \bar{\pi}^*_{m+1,j} + (2\xi + A) \bar{\pi}^*_{m+1,n+1} \\
&= \sum_{i=1}^m \sum_{j=1}^n M_{i,j} \bar{\pi}^*_{i,j} G_{i,j} + \xi(\|\boldsymbol{p}\|_1 + \|\boldsymbol{q}\|_1 - 2s - 2t) + (2\xi + A)t \\
&= \sum_{i=1}^m \sum_{j=1}^n M_{i,j} \bar{\pi}^*_{i,j} G_{i,j} + \xi(\|\boldsymbol{p}\|_1 + \|\boldsymbol{q}\|_1 - 2s) + At.
\end{aligned}
\tag{A-38}
$$

Suppose $\bar{\pi}^*_{m+1,n+1} > 0$, we next construct a solution $\gamma$ such that $\gamma_{m+1,n+1} = 0$ and leads to conflict. We randomly select a set $S = \{(i,j) | \bar{\pi}^*_{i,j} > 0, i \le m, j \le n, i \notin \mathcal{I}, j \notin \mathcal{J}\}$ and a index pair $(i_0, j_0)$ satisfying the constraints of elements in $S$, such that $\sum_{(i,j) \in S} \bar{\pi}^*_{i,j} \le t$ and $\sum_{(i,j) \in S} \bar{\pi}^*_{i,j} + \bar{\pi}^*_{i_0,j_0} > t$. In the rest part of this section, the involved $i, j$ satisfy $i \le m$ and $j \le n$. Such non-empty $S$ and

$(i_0, j_0)$ always exist, because

$$
\begin{aligned}
\mathbb{1}_m^\top (M \odot \ddot{\pi}) \mathbb{1}_n &= \sum_{i=1}^m \sum_{j=1}^n M_{i,j} \bar{\pi}_{i,j}^* = \sum_{i \in \mathcal{I}, j} \bar{\pi}_{i,j}^* + \sum_{i \notin \mathcal{I}, j \in \mathcal{J}} M_{i,j} \bar{\pi}_{i,j}^* + \sum_{i \notin \mathcal{I}, j \notin \mathcal{J}} \bar{\pi}_{i,j}^* \\
&= \sum_{i \in \mathcal{I}, j} \bar{\pi}_{i,j}^* + \sum_{i \notin \mathcal{I}, j \in \mathcal{J}} M_{i,j} \bar{\pi}_{i,j}^* + \sum_{i \notin \mathcal{I}, j \notin \mathcal{J}} \bar{\pi}_{i,j}^* \\
&= \sum_{i \in \mathcal{I}, j} \bar{\pi}_{i,j}^* + \sum_{i \notin \mathcal{I}, i' \in \mathcal{I}} M_{i,\kappa(i')} \bar{\pi}_{i,\kappa(i')}^* + \sum_{i \notin \mathcal{I}, j \notin \mathcal{J}} \bar{\pi}_{i,j}^* \qquad \text{(A-39)} \\
&\leq \sum_{i \in I} p_i + \sum_{i' \in \mathcal{I}, i \neq i'} M_{i,\kappa(i')} \bar{\pi}_{i,\kappa(i')}^* + \sum_{i \notin \mathcal{I}, j \notin \mathcal{J}} \bar{\pi}_{i,j}^* \\
&= \sum_{i \in I} p_i + \sum_{i' \in \mathcal{I}} \max\{q_{\kappa(i')} - p_{i'}, 0\} + \sum_{i \notin \mathcal{I}, j \notin \mathcal{J}} \bar{\pi}_{i,j}^*,
\end{aligned}
$$

$\mathbb{1}_m^\top (M \odot \ddot{\pi}) \mathbb{1}_n = s + t$, and $\sum_{i \in \mathcal{I}} p_i + \max\{q_{\kappa(i)} - p_i, 0\} < s$, we have $\sum_{i \notin \mathcal{I}, j \notin \mathcal{J}} \bar{\pi}_{ij}^* > t$. We now move the mass of index pairs in $S$ and $(i_0, j_0)$ to their marginal such that a total mass of $t$ is moved. Specifically, for $(i,j) \in S$, we set $\gamma_{i,j} = 0, \gamma_{i,n+1} = \bar{\pi}_{i,n+1}^* + \bar{\pi}_{i,j}^*, \gamma_{m+1,j} = \bar{\pi}_{m+1,j}^* + \bar{\pi}_{i,j}^*$. For $(i_0, j_0)$, we set $\gamma_{i_0,j_0} = \bar{\pi}_{i_0,j_0}^* - (t - \sum_{i \notin \mathcal{I}, j \notin \mathcal{J}} \bar{\pi}_{i,j}^*), \gamma_{i_0,n+1} = \bar{\pi}_{i_0,n+1}^* + (t - \sum_{i \notin \mathcal{I}, j \notin \mathcal{J}} \bar{\pi}_{i,j}^*), \gamma_{m+1,j_0} = \bar{\pi}_{m+1,j_0}^* - (t - \sum_{i \notin \mathcal{I}, j \notin \mathcal{J}} \bar{\pi}_{i,j}^*)$. For $(i,j) \notin S$, we set $\gamma_{i,j} = \bar{\pi}_{i,j}^*, \gamma_{i,n+1} = \bar{\pi}_{i,n+1}^*, \gamma_{m+1,j} = \bar{\pi}_{m+1,j}^*$. It is easy to verify that $\gamma \in \Pi(\bar{p}, \bar{q}; \bar{M})$. Similar to Eq. (A-38), we have

$$
\langle \bar{M} \odot \gamma, \bar{G} \rangle_F = \sum_{i=1}^m \sum_{j=1}^n M_{i,j} \gamma_{i,j} G_{i,j} + \xi(\|p\|_1 + \|q\|_1 - 2s). \qquad \text{(A-40)}
$$

Using the optimality of $\bar{M} \odot \pi^*$, we have

$$
\langle \bar{M} \odot \gamma, \bar{G} \rangle_F - \langle \bar{M} \odot \bar{\pi}^*, \bar{G} \rangle_F = \sum_{i=1}^m \sum_{j=1}^n M_{i,j} (\gamma_{i,j} - \bar{\pi}_{i,j}^*) G_{i,j} - At > 0. \qquad \text{(A-41)}
$$

From the definition of $\gamma$, we can see that $\gamma_{i,j} \leq \bar{\pi}_{i,j}^*$, and thus $\sum_{i=1}^m \sum_{j=1}^n M_{i,j} (\gamma_{i,j} - \bar{\pi}_{i,j}^*) G_{i,j} \leq 0$. Hence, from Eq. (A-41), we have $A < 0$, contradicting the assumption that $A > 0$. Therefore, $t = \bar{\pi}_{m+1,n+1}^* = 0$ holds.

**Step 2: show that $\ddot{\pi}$ is a feasible solution of problem in Eq. (A-31).**

We verify the constraints as follows.

(1) Since $\bar{\pi}^* \geq 0$, we have $\ddot{\pi} \geq 0$.

(2) $(\bar{M} \odot \bar{\pi}^*) \mathbb{1}_{n+1} = \begin{bmatrix} M \odot \ddot{\pi} & a \odot \bar{\pi}_{1:m,n+1}^* \\ b \odot \bar{\pi}_{m+1,1:n}^* & 0 \end{bmatrix} \begin{bmatrix} \mathbb{1}_n \\ 1 \end{bmatrix} = \begin{bmatrix} p \\ \|q\|_1 - s \end{bmatrix}$, then $(M \odot \ddot{\pi}) \mathbb{1}_n + a \odot \bar{\pi}_{1:m,n+1}^* = p$, and $(M \odot \ddot{\pi}) \mathbb{1}_n \leq p$.

(3) Similarly, from $\mathbb{1}_{m+1}^\top (\bar{M} \odot \bar{\pi}^*) = (q, \|q\|_1 - s)^\top$, we have $\mathbb{1}_m^\top (M \odot \ddot{\pi}) \leq q$.

(4) $\mathbb{1}_m^\top (M \odot \ddot{\pi}) \mathbb{1}_n = s$ holds, because $t = 0$ as in Step 1.

(5) $\forall i \in \mathcal{I}, (\bar{M} \odot \bar{\pi}^*)_{i,:} \mathbb{1}_{n+1} = (M \odot \ddot{\pi})_{i,:} \mathbb{1}_n + a_i \bar{\pi}_{i,n+1}^* = p_i$. Since $a_i = 0$, we have $(M \odot \ddot{\pi})_{1,:} \mathbb{1}_n = p_i$.

(6) $\forall j \in \mathcal{J}, \mathbb{1}_{m+1}^\top (\bar{M} \odot \bar{\pi}^*)_{:,j} = \mathbb{1}_m^\top (M \odot \ddot{\pi})_{:,j} + b_j \bar{\pi}_{m+1,j}^* = q_j$. Since $b_j = 0$, we have $\mathbb{1}_m^\top (M \odot \ddot{\pi})_{:,j} = q_j$.

Therefore, we have $\ddot{\pi} \in \Pi^s(p, q; M)$, and $\ddot{\pi}$ is a feasible solution of problem in Eq. (A-31).

**Step 3: show that $M \odot \ddot{\pi}$ is the optimal transport plan of problem in Eq. (A-31).**

Suppose there exist a transport plan $M \odot \gamma$ with $\gamma \in \Pi^s(p, q; M)$ such that

$$
\sum_{i=1}^m \sum_{j=1}^n M_{i,j} \gamma_{i,j} G_{i,j} < \sum_{i=1}^m \sum_{j=1}^n M_{i,j} \ddot{\pi}_{i,j} G_{i,j}.
$$

We construct $\bar{\gamma}$ as follows. For $i \leq m, j \leq n$, $\bar{\gamma}_{i,j} = \gamma_{i,j}$. $\bar{\gamma}_{i,n+1} = p_i - \sum_{j=1}^{n} \gamma_{i,j}, \forall i \leq m$. $\bar{\gamma}_{m+1,j} = q_j - \sum_{i=1}^{n} \gamma_{i,j}, \forall j \leq n$. $\bar{\gamma}_{m+1,n+1} = 0$. Easily, we can verify that $\bar{\gamma}$ is in $\Pi(\bar{p}, \bar{q}; \bar{M})$. Meanwhile,

$$
\begin{aligned}
\langle \bar{M} \odot \bar{\gamma}, \bar{G} \rangle_F &= \sum_{i=1}^{m} \sum_{j=1}^{n} M_{i,j} \gamma_{i,j} C_{i,j} + \xi(\|p\|_1 + \|q\|_1 - 2s) \\
&< \sum_{i=1}^{m} \sum_{j=1}^{n} M_{i,j} \ddot{\pi}_{i,j} G_{i,j} + \xi(\|p\|_1 + \|q\|_1 - 2s) \\
&= \langle \bar{M} \odot \bar{\pi}^*, \bar{G} \rangle_F.
\end{aligned}
\tag{A-42}
$$

This contradicts the fact that $\bar{M} \odot \bar{\pi}^*$ is the optimal transport plan of problem $\min_{\bar{\pi} \in \Pi(\bar{p}, \bar{q}; \bar{M})} \langle \bar{M} \odot \bar{\pi}, \bar{G} \rangle_F$. Therefore, $M \odot \ddot{\pi}$ is the optimal transport plan of problem in Eq. (A-31). $\qquad \square$

# B  Additional Experimental Details and Results

## B.1  Toy Experiment for Evaluating Partial-KPG-RL model

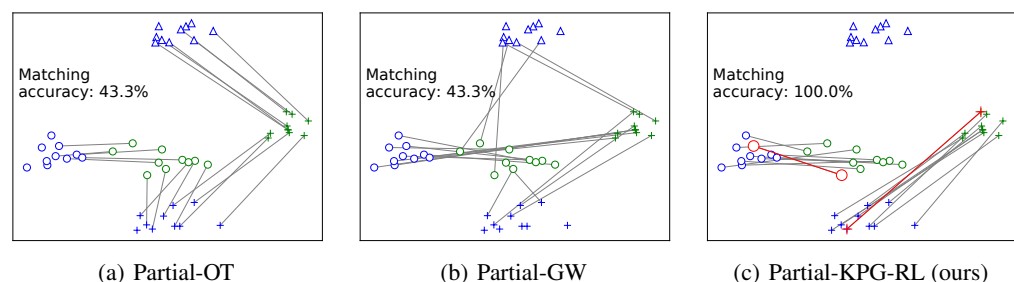

| (a) Partial-OT | (b) Partial-GW | (c) Partial-KPG-RL (ours) |

Figure A-3: Matching produced by (a) partial-OT model, (b) partial-GW model, and (c) our proposed partial-KPG-RL model.

Figure A-3 illustrates the toy data experiment for evaluating the partial-KPG-RL model. In Fig. A-3, the source (blue) and target (green) distributions are Gaussian mixtures. The source (resp. target) distribution is composed of three (resp. two) distinct Gaussian components indicated by different shapes where the same shapes indicate the points of the same class. When conducting OT, the source class data represented by "$\triangle$" should not be transported. In Figs. A-3(a) and A-3(b), we can observe that both the partial-OT model (defined in Eq. (2) in the paper) and the partial-GW model [20] wrongly transport some source points of class "$\triangle$" to target domain and lead to low matching accuracy. With the guidance of a few keypoints (red pairs), our proposed partial-KPG-RL model does not transport the source points of class "$\triangle$" to target domain and apparently improves the matching accuracy as in Fig. A-3(c).

## B.2  Additional Experimental Details and Results for Open-set HDA

**More experimental details.**   In open-set HDA, we are given a large amount of labeled source domain data $\{(x_i, t_i)\}_{i=1}^{m}$, a few labeled target domain data $\{y_j, \bar{t}_j\}_{j=1}^{n_l}$, and a large number of unlabeled target domain data $\{y_j\}_{j=n_l+1}^{n}$. The fraction of unknown class data is $\eta$. To apply the partial-KPG-RL model defined in Eq. (13) in the paper to open-set HDA, for each labeled target domain data, we take its corresponding source class center to construct a keypoint pair. We then resample the source domain data such that the total number of resampled source domain data and the source keypoints is $m' = (1 - \eta)n$. We define the source distribution as $p = \frac{1-\eta}{m'}(\sum_{j=1}^{n_l} \delta_{c_j} + \sum_{j=n_l+1}^{m'} \delta_{x'_j})$, where $x'_j$ is a resmapled source domain sample and $c_j$ is the source class center corresponding to the target labeled sample $y_j$. The target distribution is defined as $q = \frac{1}{n} \sum_{j=1}^{n} \delta_{y_j}$.

The partial-KPG-RL model is conducted to transport mass from $\boldsymbol{p}$ to $\boldsymbol{q}$ with $s = 1-\eta$. After transport, the $\eta$-fraction unlabeled target data receiving smallest mass from source domain are detected as unknown class and the rest unlabeled target data are taken as common class ones. Finally, we train the kernel SVM on the transported source domain data and labeled target domain data to classify the unlabeled target domain common class data.

Table A-1: Results on Office-31 for open-set HDA with unknown $\eta$. $\hat{\eta}$ is the estimate of $\eta$ (the true $\eta = 0.67$).

| Method | A→A ($\hat{\eta} = 0.57$) | | | A→D ($\hat{\eta} = 0.48$) | | | A→W ($\hat{\eta} = 0.62$) | | | D→A ($\hat{\eta} = 0.57$) | | | D→D ($\hat{\eta} = 0.48$) | | |
|---|---|---|---|---|---|---|---|---|---|---|---|---|---|---|---|
| | OS* | UNK | HOS | OS* | UNK | HOS | OS* | UNK | HOS | OS* | UNK | HOS | OS* | UNK | HOS |
| Baseline | 38.2 | 61.9 | 47.2 | 20.0 | **69.3** | 31.0 | 28.2 | **80.1** | 41.7 | 38.2 | 61.9 | 47.2 | 20.0 | **69.3** | 31.0 |
| **Partial-KPG-RL** | **49.1** | **70.1** | **57.8** | **61.8** | 59.3 | **60.5** | **54.5** | 73.2 | **62.5** | **59.1** | **73.6** | **65.5** | **83.6** | 66.7 | **74.2** |

| Method | D→W ($\hat{\eta} = 0.62$) | | | W→A ($\hat{\eta} = 0.57$) | | | W→D ($\hat{\eta} = 0.48$) | | | W→W ($\hat{\eta} = 0.62$) | | | **Avg** | | |
|---|---|---|---|---|---|---|---|---|---|---|---|---|---|---|---|
| | OS* | UNK | HOS | OS* | UNK | HOS | OS* | UNK | HOS | OS* | UNK | HOS | OS* | UNK | HOS |
| Baseline | 28.2 | 80.1 | 41.7 | 38.2 | 61.9 | 47.2 | 20.0 | **69.3** | 31.0 | 28.2 | 80.1 | 41.7 | 28.8 | 70.4 | 40.0 |
| **Partial-KPG-RL** | **78.2** | **87.4** | **82.6** | **60.9** | **74.5** | **67.0** | **81.8** | 66.7 | **73.5** | **78.2** | **87.0** | **82.4** | **67.5** | **73.2** | **69.5** |

**Results for open-set HDA with unknown $\eta$.** For the more practical open-set HDA setting that $\eta$ is unknown, researchers can design methods to estimate $\eta$ and then apply our method using the estimate of $\eta$, or take $\eta$ as a hyper-parameter and design methods to tune it. We directly use the positive-unlabeled learning [63] method [64] to estimate the fraction of common class data among the target domain unlabeled data, by taking the labeled target data as positive samples. The results of different methods for open-set HDA using the estimate $\hat{\eta}$ of $\eta$ are given in Table A-1. According to Table A-1, the positive transfer is achieved by our method. We can see that $\hat{\eta}$ in all tasks is lower than the true $\eta$, implying that less unknown class samples are detected. Correspondingly, the UNK value (73.2%) achieved by partial-KPG-RL using $\hat{\eta}$ in Table A-1 is smaller than that (83.5%) using $\eta$ in Table 3 in the paper. Surprisingly, the OS* value (67.5%) of partial-KPG-RL in Table A-1 is higher than that (59.7%) in Table 3 in the paper. As a balance, the HOS value (69.5%) achieved by partial-KPG-RL using $\hat{\eta}$ is similar to the HOS value (69.1%) of partial-KPG-RL using the true $\eta$.

In the following Table A-2, we take $\eta$ as a hyper-parameter and show the average HOS achieved by partial-KPG-RL using varying magnitude of $\eta$. It is observed that the average HOS is stable to $\eta$ in a relatively large range of $[0.50, 0.80]$.

Table A-2: Average HOS of partial-KPG-RL using varying magnitude of $\eta$ (the unknown true value of $\eta$ is 0.67).

| $\eta$ | 0.50 | 0.55 | 0.60 | 0.65 | 0.70 | 0.75 | 0.80 |
|---|---|---|---|---|---|---|---|
| Average HOS | 67.2 | 69.2 | 69.9 | 68.7 | 69.2 | 68.5 | 65.5 |

## B.3 Application in Deep Unsupervised Domain Adaptation

In this section, we apply our method to deep unsupervised domain adaptation where the mini-batch-based implementation is required. The main challenge is that some of the samples in the mini-batch may not be matched. For instance, the categories of some samples in the source mini-batch may not be present in the target mini-batch, and thus these source samples should not be transported/matched. Inspired by [65] that uses partial OT over the mini-batch data to implement deepJDOT [66], we use our partial KPG-RL-KP model to partially match the mini-batch data in the training of the deep network. The partial KPG-RL-KP model is modified from Eq. (13) by replacing $G$ by $\alpha C + (1-\alpha)G$. As an experimental example, we apply the partial KPG-RL-KP to the unsupervised domain adaptation experiment on the Office-Home dataset [67]. We take the source and target class centers of the same class as a keypoint pair. The centers are online updated by exponential moving average in training, same as in [68]. We use the pseudo labels of target data to update the target class centers, due to the

lack of target labels. The protocol is the same as that in [65] . The batch size is set to 65 and the total transport mass ($s$ in Eq. (13)) is set to 0.6, which are the same as those in [65]. The results are reported in the following Table A-3.

Table A-3: Results for unsupervised domain adaptation. "A", "C", "P", and "R" are the domains of "Art", "Clipart", "Product", and "RealWorld" in Office-Home dataset.

| Method | A→C | A→P | A→R | C→A | C→P | C→R | P→A | P→C | P→R | R→A | R→C | R→P | Avg |
|---|---|---|---|---|---|---|---|---|---|---|---|---|---|
| ROT [23] | 47.20 | 71.80 | 76.40 | 58.60 | 68.10 | 70.20 | 56.50 | 45.00 | 75.80 | 69.40 | 52.10 | 80.60 | 64.30 |
| m-OT [66] | 51.75 | 70.01 | 75.79 | 59.60 | 66.46 | 70.07 | 57.60 | 47.88 | 75.29 | 66.82 | 55.71 | 78.11 | 64.59 |
| m-UOT [69] | 54.99 | **74.45** | **80.78** | 65.66 | 74.93 | 74.91 | 64.70 | 53.42 | 80.01 | 74.58 | 59.88 | 83.73 | 70.17 |
| m-POT [65] | 55.65 | 73.80 | 80.76 | **66.34** | 74.88 | **76.16** | 64.46 | 53.38 | **80.60** | 74.55 | 59.71 | 83.81 | 70.34 |
| **m-KPG-RL-KP** | 52.13 | 63.65 | 74.53 | 61.12 | 67.84 | 67.88 | 59.84 | 52.93 | 76.90 | 71.92 | 59.21 | 82.55 | 65.88 |
| **m-PKPG-RL-KP** | **57.96** | **74.45** | 78.75 | 66.30 | **75.22** | 74.39 | **66.87** | **58.47** | 80.47 | **75.15** | **61.15** | **84.23** | **71.12** |

In Table A-3, ROT [23] is a robust OT method. m-OT is the direct mini-batch implementation of deepJDOT [66]. m-UOT [69] and m-POT [65] are respectively unbalanced deepJDOT and partial deepJDOT on mini-batch data. m-KPG-RL-KP is the direct mini-batch implementation of our KPG-RL-KP model. m-PKPG-RL-KP is the mini-batch implementation of our partial KPG-RL-KP model. We can see that by partially matching the samples in the mini-batches, m-KPG-RL-KP outperforms m-KPG-RL-KP by a margin of 6.24%. Our partial KPG-RL-KP (m-PKPG-RL-KP) outperforms partial DeepJDOT (m-POT) by 0.68%, indicating that using partial matching, our approach is effective for unsupervised domain adaptation under mini-batch implementation.

## B.4 Additional Details for HDA Experiments

**Kernel SVM.** In the kernel SVM, we use the radial basis function kernel $k(x, y) = \exp(-\gamma \|x - y\|^2)$, where $\gamma$ is set to the reciprocal of the feature dimension. We use the scikit-learn packadge of python to implement it by simply running the following codes:

```
clf = SVC(gamma='auto')
clf.fit(feat_train,label_train)
```

**Barycentric mapping.** The barycentric mapping is defined as follows. Given the transport plan $\pi \in \Sigma_{m \times n}$ and source data point $x_{i_0}$, the barycentric mapping [49] is defined as $B_\pi(x_{i_0}) = \arg\min_y \sum_{j=1}^n \pi_{i_0,j} c(x_{i_0}, y_j)$. Since $c$ is the squared $L_2$-distance in our paper, $B_\pi(x_{i_0})$ has closed-form expression of

$$B_\pi(x_{i_0}) = \frac{1}{\sum_{j=1}^n \pi_{i_0,j}} \sum_{j=1}^n \pi_{i_0,j} y_j. \tag{A-43}$$

## B.5 Additional Ablation Studies

**Matching accuracy on real data in HDA application.** We compare the matching accuracy of different OT models on Office-31 dataset in Table A-4. To compute the matching accuracy, for each transported source data point, we find its nearest neighbor among the target data points to construct a matched pair. If the two points in a pair have the same class labels, the matching is correct, otherwise the matching is incorrect. The matching accuracy is the ratio of correctly matched pairs. In Table A-4, we can see that without the guidance of keypoints, the matching accuracy of GW model is less than 3%. SGW improves the matching accuracy of GW. Our proposed KPG-RL and KPG-RL-GW models achieve better matching accuracy than SGW.

Table A-4: Matching accuracy of different OT models on Office-31 for HDA tasks.

| OT models | A→A | A→D | A→W | D→A | D→D | D→W | W→A | W→D | W→W | Avg |
|---|---|---|---|---|---|---|---|---|---|---|
| GW [27] | 2.5 | 0.7 | 1.4 | 2.9 | 1.8 | 1.8 | 2.5 | 0.4 | 0.4 | 1.6 |
| SGW [37] | 43.4 | 61.7 | 64.3 | 43.8 | 73.1 | 68.9 | 43.4 | 72.5 | 72.7 | 60.4 |
| **KPG-RL** | 48.7 | **67.7** | 66.1 | **50.7** | **86.8** | **81.0** | 51.0 | **82.9** | **83.3** | **68.7** |
| **KPG-RL-GW** | **50.3** | 67.4 | **66.2** | 49.7 | 86.5 | **81.0** | **51.9** | **82.9** | 82.6 | **68.7** |

**Comparison of different choices for $d$.**  Since $R_k^s$ and $R_l^t$ are in the probability simplex, it is reasonable to measure their difference by a distribution divergence/distance. The widely used distribution divergences/distances include the KL-divergence, JS-divergence, and Wasserstein distance. The KL-divergence is not symmetric, so we need to determine the order of inputs. For the Wasserstein distance, one should define the ground metric first. A possible strategy is to set the ground metric to 0 if the two keypoints are paired, otherwise 1. Such a ground metric makes the Wasserstein distance equal to the $L_1$-distance. In this work, $d$ is taken as the JS-divergence. We compare the performance of different choices of $d$ in the experiment of HDA on Office-31, as in Table A-5.

Table A-5: Results of different choices of $d$ in HDA experiment on Office-31.

| Choices of $d$ | A→A | A→D | A→W | D→A | D→D | D→W | W→A | W→D | W→W | Avg |
|---|---|---|---|---|---|---|---|---|---|---|
| KL-ST | 59.0 | 89.7 | **83.6** | 56.8 | 95.2 | **89.0** | 57.7 | 93.6 | 88.1 | 79.2 |
| KL-TS | 58.1 | 89.0 | 82.3 | 54.2 | 93.9 | 88.1 | 54.2 | 93.2 | **89.4** | 78.0 |
| $L_1$-distance | 57.4 | 85.8 | 79.0 | 58.0 | 85.8 | 82.9 | 58.4 | 92.6 | 83.6 | 75.9 |
| $L_2$-distance | 52.3 | 85.8 | 81.3 | 53.2 | 91.3 | 82.3 | 52.6 | 90.3 | 82.9 | 74.7 |
| GW | 42.0 | 71.6 | 70.0 | 41.6 | 71.0 | 69.4 | 42.3 | 71.3 | 70.0 | 61.0 |
| JS | **60.0** | **91.6** | **83.6** | **57.4** | **95.8** | 87.7 | **59.1** | **95.2** | 88.4 | **79.9** |

In Table A-5, KL-ST and KL-TS denote the KL-divergence $KL(R_k^s, R_l^t)$ and $KL(R_l^t, R_k^s)$ respectively. GW is the Gromov-Wasserstein distance between $R_k^s$ and $R_l^t$ where the source/target cost is taken as the $L_2$-distance of source/target keypoints. We find that the JS-divergence achieves the best performance, compared with KL-ST, KL-TS, $L_1$-distance, $L_2$-distance, and Gromov-Wasserstein.

**Results of KPG-RL without using the guiding matrix $G$.**  The guiding matrix is the core to impose the relation preservation. We below show in Table A-6 the results for our KPG-RL without using the guiding matrix $G$, *i.e.*, $L_{kpg}(\pi) = \langle M, \pi \rangle_F$.

Table A-6: Results for different definitions of $L_{kpg}(\pi)$ in HDA experiment.

| Definition of $L_{kpg}(\pi)$ | $\langle M, \pi \rangle_F$ | $\langle M \odot \pi, G \rangle_F$ |
|---|---|---|
| KPG-RL | 60.7 | 79.9 |
| KPG-RL-GW | 60.6 | 79.6 |

We can see that without $G$, both the results of KPG-RL model and KPG-RL-GW model decrease. This may be because $L_{kpg}(\pi) = \langle M, \pi \rangle_F$ may not well impose the guidance of keypoints, since it does not model the "relation" of each point to keypoints.

**Sensitivity to source keypoints.**  In the experiments of the paper, the source keypoints are taken as the source class centers. To study the sensitivity to the location of source keypoints, we randomly sample one data point from each class as a keypoint to construct the source keypoints. We run the experiments with five different samplings for constructing the source keypoints (these five runs are denoted as S1, S2, S3, S4, S5 respectively). The results are reported in the Table A-7. We can see that using the class center as the keypoints achieves the best results, compared with randomly sampling one data point per class as the keypoints. This may be because the class centers are estimated using all the data of each class, and these centers can better represent each class than a randomly sampled data point of each class.

Table A-7: Results for different locations of source keypoints.

| S1 | S2 | S3 | S4 | S5 | Centers |
|------|------|------|------|------|---------|
| 76.8 | 77.5 | 78.2 | 77.8 | 76.9 | 79.9 |

We next study the sensitivity to the number of source keypoints, of which the results are reported in Table A-8. In this experiment, we randomly sample 3/5/7/9 samples (keypoints) or use all the source samples (keypoints) for each class in the source domain to compute the source class centers, which are paired with labeled target samples for constructing the keypoint pairs. The results in Table A-8 show that as the number of source keypoints increases, the accuracy gradually increases. The best result is obtained when all source samples are used to compute the class centers.

Table A-8: Results for different numbers of source keypoints.

| Number | 3 | 5 | 7 | 9 | All |
|----------|------|------|------|------|------|
| Accuracy | 78.4 | 79.2 | 79.6 | 79.8 | 79.9 |

**On defining keypoints in other practical applications.**  According to the results in Table A-7, the class centers are better to be the keypoints than the randomly selected samples. For other practical applications, there may not be "class labels" available. We could first cluster the points and then annotate the points near to the center of the clusters as the keypoints.

**Time and memory cost.**  We report the memory and time cost of KPG-RL with different sizes of the guiding matrix $G$ in the bottom row of Tables A-9 and A-10 respectively. For comparisons, we also report the memory and time cost of the Kantorovich Problem (KP). KP needs to calculate the pair-wised cost matrix $C$, as in Eq. (1). KPG-RL calculates the relation score, and then computes the guiding matrix $G$. Since we have deduced Sinkhorn's algorithm for solving KPG-RL, we solve both KP and KPG-RL using Sinkhorn's algorithm with $\epsilon = 0.005$. Table A-9 shows that KPG-RL costs a slightly larger memory than KP. Table A-10 shows the computational time for solving KPG-RL and KP problems. In the experiment on Office-31, the maximum memory of $G$ is 38M, and the peak memory of the running process is 780M.

Table A-9: Peak memory for computing KP and KPG-RL.

| Size ($m \times n$) of $C$/$G$ | $500 \times 500$ | $1000 \times 1000$ | $2000 \times 2000$ |
|--------------------------------|------------------|--------------------|--------------------|
| KP | 201M | 218M | 330M |
| KPG-RL | 207M | 232M | 378M |

Table A-10: Time cost for computing KP and KPG-RL.

| Size ($m \times n$) of $C$/$G$ | $500 \times 500$ | $1000 \times 1000$ | $2000 \times 2000$ |
|--------------------------------|------------------|--------------------|--------------------|
| KP | 6.9s | 27.7s | 60.1s |
| KPG-RL | 10.8s | 42.1s | 76.5s |

**Sensitivity to hyper-parameters.**  We show the sensitivity of our method to hyper-parameters $\tau, \tau'$ in Table A-11, $\epsilon$ in Table A-12, and $\alpha$ in Table A-13. $\epsilon$ is the the coefficient of entropy regularization. $\tau$ and $\tau'$ are used to define the relation in Eqs. (7) and (8) in the paper. We set $\tau = \rho \max_{i,j}\{C_{i,j}^s\}$ and $\tau' = \rho \max_{i,j}\{C_{i,j}^t\}$. We then show the results with varying values of $\rho$. It can be observed that the best value of $\alpha$ is 0.4 in this task, and the results are relatively stable when $\alpha$ ranges in [0.2, 0.5].

Table A-11: Sensitivity of KPG-RL to hyper-parameters $\tau$ and $\tau'$ in HDA task A→W. We set $\tau = \rho \max_{i,j}\{C^s_{i,j}\}$ and $\tau' = \rho \max_{i,j}\{C^t_{i,j}\}$. We then show the results with varying values of $\rho$.

| $\rho$ | 0.05 | 0.07 | 0.09 | 0.1 | 0.2 | 0.3 | 0.4 | 0.5 |
|---|---|---|---|---|---|---|---|---|
| Accuracy | 82.3 | 83.2 | 83.2 | 83.6 | 83.2 | 82.9 | 82.6 | 82.6 |

Table A-12: Sensitivity of KPG-RL to hyper-parameter $\epsilon$ in HDA task A→W.

| $\epsilon$ | 0.0001 | 0.0005 | 0.001 | 0.005 | 0.01 | 0.05 | 0.1 | 1 |
|---|---|---|---|---|---|---|---|---|
| Accuracy | 83.2 | 83.2 | 83.3 | 83.6 | 82.3 | 76.5 | 74.8 | 71.0 |

Table A-13: Sensitivity of KPG-RL-GW to hyper-parameter $\alpha$ in HDA task A→W.

| $\alpha$ | 0.9 | 0.8 | 0.7 | 0.6 | 0.5 | 0.4 | 0.3 | 0.2 | 0.1 |
|---|---|---|---|---|---|---|---|---|---|
| Accuracy | 74.3 | 78.1 | 81.5 | 82.9 | 84.2 | 84.5 | 84.0 | 84.0 | 83.7 |