# OpenReview forum: "Keypoint-Guided Optimal Transport with Applications in Heterogeneous Domain Adaptation"
_NeurIPS.cc/2022/Conference — NeurIPS 2022 Accept_

### Official Review · Reviewer_BBx7 · 2022-06-27

**Rating:** 7
**Confidence:** 3
**Soundness:** 3 good
**Presentation:** 3 good
**Contribution:** 3 good

**Summary:**

This paper presents the KeyPoint-Guided model by ReLation preservation (KPG-RL), which allows to combine annotation information into many popular optimal transport (OT) models. The authors show the very competitive performance when integrating KPG-RL into the classical OT models, for the heterogeneous domain adaptation task.

**Questions:**

- I find the idea of introducing $G$ interesting and smart. So I am curious to see how would KPG-RL perform without $G$?, i.e. what if we only use $L_{kpg}(\pi) = \langle M, \pi \rangle_F$?

- Each entry in the matrix $G$ is the Jensen-Shannon (JS) divergence between two probability vectors. How does the choice of measure of similarity impact the performance of KPG-RL? Why JS divergence but not the KL divergence, or $L_2, L_1$ distances, or even Wasserstein distance (which is not computationally expensive in this case).

- Do the authors have the motivation to prefer the KPG-RL to the fused GW model above?

- What is the range of $\alpha$ in the equations $11$ and $12$? Is it $[0,1]$? Is it tuned during the training?

- The alignment in the Fig 4 is not very visible.

- In the experiment 5.2, how is the target data transformed for the GW model?

- In the Appendix, please provide more details on the barycentric mapping in $[48]$.

**Ethics Review Area:**

["I don’t know"]

**Limitations:**

No, the authors do not discuss these points.

**Strengths And Weaknesses:**

The paper is well written and the authors have well illustrated the advantage and motivation of KPG-RL over the traditional GW and KP models.

However, I would say that the main contribution is mostly on the methodology, so more theoritical understand would be desirable. Apart from a few theoritical results which help validating the intuition and motivation, I think it is also necessary to convince that KPG-RL is a reliable divergence/metric (even though the empirical evidence suggests that it is). More precisely, given prior correct matching of keypoint pairs,

- In the setting of OT distance, where points lie on the same ground space, do we have that $L_{kpg} = 0$ (or $\text{KPG-RL-KP} = 0$) is equivalent to the equality of two reference measures $p$ and $q$?

- When KPG-RL is incorporated in the GW model (i.e. KPG-RL-GW), does it still preserve the isomorphism?, i.e. when the two graphs are isomorphic (in the usual sense of GW distance), do we have $\text{KPG-RL-GW} = 0$, and vice versa?

- Can we show that KPG-RL or any of its variations (GW and LP) defines a proper metric?

Moreover, while the usual GW model performs very poorly because it can not incorporate prior information, the fused GW model $[35]$ can and should be considered as competing method. For example, one can consider the fused GW model of the form: $\min_{\pi \in \Pi(p,q)}\alpha L_{gw}(\pi) + (1 - \alpha) L_w(\pi)$, where either $L_w(\pi) = \langle M, \pi \rangle$, or $L_w(\pi) = \langle M \odot G, \pi \rangle$.

---

> ### Author Response · Authors · 2022-08-02
> **Response to Reviewer BBx7 Part (3/3)**
>
> **Q7: Do KPG-RL, KPG-RL-KP, or KPG-RL-GW provide divergences/metrics?**
>
> Thanks for the suggestion. Given the annotated correct matched keypoint pairs, we have proved that the KPG-RL-KP provides a proper metric and the KPG-RL-GW provides a divergence.
>
> * For $\min_{\pi\in\Pi(p,q,M)}{L_{kpg}(\pi)}=0$, we only have the equality of source and target relation scores (defined in Eqs. (7) and (8)) rather than the equality of source and target points, because the source and target keypoints could be different. Therefore, $\min_{\pi\in\Pi(p,q,M)}{L_{kpg}(\pi)}=0$ does not imply $p=q$.
> * When the points lie in the same ground space, the "correct" matched keypoint pairs implies that if $p=q$, the paired keypoints must be equal, i.e., for any $(i,j)\in\mathcal{K}$, we have $x_i=y_j$. In such a case, $p=q$ if and only if $\min_{\pi\in\Pi(p,q;M)}{\langle M\odot\pi,\alpha C + (1-\alpha)G\rangle}=0$, which means the KPG-RL-KP provides a divergence. The proof follows the proof idea of the Wasserstein distance (Proposition 2.2 in [r1]).
> * For the KPG-RL-GW model, the "correct" matched keypoint pairs implies that if there is an isometric bijection $\sigma$ between two graphs (modeled as distributions $p$ and $q$), i.e., isomorphism, we have that $\sigma$ maps the source keypoint to its paired target keypoint. In this case, the two graphs are isomorphic if and only if $\min_{\pi\in\Pi(p,q;M)}\\{\alpha L_{gw}(M\odot\pi) + (1-\alpha)L_{kpg}(\pi)\\}=0$. The proof follows the proof idea of Theorem 3.2 in the paper of fused GW [35].
> * When the points lie in the same ground space, the KPG-RL-KP model provides a proper metric, if both $c$ and $d$ are distances. The symmetry is easy to verify, because both $C$ and $G$ are symmetric. The triangle inequality follows the proof idea of the Wasserstein distance (Proposition 2.2 in [r1]).
>
> We will include a Proposition for describing these properties and detailed proofs in Appendix A.
>
> [r1] Peyré G, Cuturi M. Computational optimal transport: With applications to data science[J]. Foundations and Trends® in Machine Learning, 2019, 11(5-6): 355-607.

---

> > ### Comment · Reviewer_BBx7 · 2022-08-03
> > **Response to Authors**
> >
> > I thank the authors for their response.
> >
> > The authors have well addressed all of my questions. As long as the authors include all additional clarification and details in the final version, I believe this paper will be a very good and solid work.
> >
> > I only have one minor concern, where I believe the parameter $\alpha$ should also be tuned, rather than being fixed. Nevertherless, the current performance of KPG is already very competitive, so the tuning might not help much.
> >
> > However, this minor concern does not prevent me to increase the score to 7.

---

> > > ### Author Response · Authors · 2022-08-05
> > > **Response to Reviewer BBx7**
> > >
> > > Thanks for the comments again. We investigate the effect of $\alpha$ in the following Table R4-5.
> > >
> > > Table R4-5. Results of KPG-RL-GW with varying values of $\alpha$ in the HDA task A$\rightarrow$W.
> > >
> > > | $\alpha$ | 0.9  | 0.8  | 0.7  | 0.6  | 0.5  | 0.4  | 0.3  | 0.2  | 0.1  |
> > > | -------- | ---- | ---- | ---- | ---- | ---- | ---- | ---- | ---- | ---- |
> > > | Accuracy | 74.3 | 78.1 | 81.5 | 82.9 | 84.2 | 84.5 | 84.0 | 84.0 | 83.7 |
> > >
> > > From Table R4-5, it can be observed that the best value of $\alpha$ is 0.4 in this task, and the results are relatively stable when $\alpha$ ranges in [0.2, 0.5]. We will include this experiment in Appendix B.

---

> > > > ### Comment · Reviewer_BBx7 · 2022-08-05
> > > > **Response to Authors**
> > > >
> > > > I thank the authors for their response.
> > > >
> > > > I appreciate the effort.
> > > > As said before, I am happy to increase the score to 7.

---

> ### Author Response · Authors · 2022-08-02
> **Response to Reviewer BBx7 Part (2/3)**
>
> **Q3: Motivation to prefer KPG-RL to fused GW model of $\min_{\pi\in\Pi(p,q)}\\{\alpha L_{gw}(\pi) + (1-\alpha) L_w(\pi)\\}$ with $L_w(\pi)=\langle M, \pi \rangle_F$ or $L_w(\pi)=\langle M\odot G, \pi \rangle_F$.**
>
> The KPG-RL models the guidance of keypoints to the other points in OT by a mask-based constraint on the transport plan to enforce the matching of keypoints and preserving the relation of each point to keypoints.  While the above fused GW models may not enforce the matching of keypoints. Table R4-3 implies that the results of the above defined fused GW models are lower than the result of KPG-RL model.
>
> Table R4-3. Results of KPG-RL and fused GW models for HDA on Office-31.
>
> | Methods  | KPG-RL | Fused GW (w/ $L_w(\pi)=\langle M, \pi \rangle_F$) | Fused GW (w/ $L_w(\pi)=\langle M\odot G, \pi \rangle_F$) |
> | -------- | :------: | :-------------------------------------------------: | :--------------------------------------------------------: |
> | Accuracy | 79.9   | 25.9                                              | 73.2                                                     |
>
> From the computation point of view,  the fused-GW models are non-convex quadratic programs, while KPG-RL is a linear program. Table R4-4 shows that the computation of the KPG-RL takes less time than that of fused GW models. Another experimental finding is that the definition of $L_{kpg}(\pi)$ affects the convergence speed of the fused GW.
>
> Table R4-4. Time cost for computing KPG-RL and fused GW models in the task A$\rightarrow$A of HDA experiment on Office-31.
>
> | Methods | KPG-RL | Fused GW (w/ $L_w(\pi)=\langle M, \pi \rangle_F$) | Fused GW (w/ $L_w(\pi)=\langle M\odot G, \pi \rangle_F$) |
> | ------- | :------: | :-------------------------------------------------: | :--------------------------------------------------------: |
> | Time    | 6.7s   | 98.1s                                             | 25.6s                                                    |
>
> **Q4: Range and set of $\alpha$.**
>
> Thanks for this question. The range of $\alpha$ is $(0,1)$. We simply set $\alpha$ to 0.5 throughout this paper.  This will be added in the revised paper.
>
> **Q5: Clarifying that the target data are not transformed for the GW model.**
>
> We use GW to learn the transport plan between source and target domain data. Then, we transport the source data using the barycentric mapping. Finally, we train the classifier on the transported source data with labels and labeled target data. We will include these details for GW in Section 5.2 in the revised paper.
>
> **Q6: Making Fig. 4 clearer and including the barycentric mapping in Appendix.**
>
> Thanks for the suggestions. We will update Fig. 4. The barycentric mapping is defined as follows. Given the transport plan $\pi\in\Sigma_{m\times n}$ and source data point $x_{i_0}$, the barycentric mapping [48] is defined as $B_{\pi}(x_{i_0})=\arg\min_y{\sum_{j=1}^n\pi_{i_0,j}c(x_{i_0},y_j)}$. Since $c$ is the squared $L_2$-distance in our paper, $B_{\pi}(x_{i_0})$ has closed-form expression of $B_{\pi}(x_{i_0})=\frac{1}{\sum_{j=1}^n\pi_{i_0,j}}\sum_{j=1}^n\pi_{i_0,j}y_j$. As suggested, we will include the barycentric mapping in Appendix B.

---

> ### Author Response · Authors · 2022-08-02
> **] Response to Reviewer BBx7 Part (1/3)**
>
> We thank the reviewer for the comments and suggestions. We will revise our paper accordingly.
>
> **Q1: Results of KPG-RL without $G$ and the fused model.**
>
> The results for $L_{kpg}(\pi)=\langle M, \pi \rangle_F$  and  $L_{kpg}(\pi)=\langle M\odot\pi,G \rangle_F$ are as follows.
>
> Table R4-1. Results for different definitions of $L_{kpg}(\pi)$ in HDA experiment.
>
> | Definition of  $L_{kpg}(\pi)$ | $\langle M, \pi \rangle_F$ | $\langle M\odot\pi,G \rangle_F$ |
> | ----------------------------- | :--------------------------: | :-------------------------------: |
> | KPG-RL                        | 60.7                       | 79.9                            |
> | KPG-RL-GW                     | 60.6                       | 79.6                            |
>
> We can see that without $G$, both the results of KPG-RL model and KPG-RL-GW model decrease. This may be because $L_{kpg}(\pi)=\langle M, \pi \rangle_F$ may not well impose the guidance of keypoints, since it does not model the "relation" of each point to keypoints. We will include this experiment in Appendix B.
>
> **Q2: Comparison of different choices for $d$.**
>
> Since $R_k^s$ and $R_l^t$ are in the probability simplex, it is reasonable to measure their difference by a distribution divergence/distance. The widely used distribution divergences/distances include the KL-divergence, JS-divergence, and Wasserstein distance. The KL-divergence is not symmetric, so we need to determine the order of inputs. For the Wasserstein distance, one should define the ground metric first. A possible strategy is to set the ground metric to 0 if the two keypoints are paired, otherwise 1.  Such a ground metric makes the Wasserstein distance equal to the $L_1$-distance. In this work, $d$ is taken as the JS-divergence. We compare the performance of different choices of $d$ in the experiment of HDA on Office-31, as in Table R4-2.
>
> Table R4-2. Results of different choices of $d$ in HDA experiment on Office-31.
>
> | Choices of $d$ | A$\rightarrow$A | A$\rightarrow$D | A$\rightarrow$W | D$\rightarrow$A | D$\rightarrow$D | D$\rightarrow$W | W$\rightarrow$A | W$\rightarrow$D | W$\rightarrow$W | Avg      |
> | -------------- | :---------------: | :---------------: |:---------------: | :---------------: | :---------------: |:---------------: | :---------------: |:---------------: |:---------------: | :---------------: |
> | KL-ST          | 59.0            | 89.7            | **83.6**        | 56.8            | 95.2            | **89.0**        | 57.7            | 93.6            | 88.1            | 79.2     |
> | KL-TS          | 58.1            | 89.0            | 82.3            | 54.2            | 93.9            | 88.1            | 54.2            | 93.2            | **89.4**        | 78.0     |
> | $L_1$-distance | 57.4            | 85.8            | 79.0            | 58.0            | 85.8            | 82.9            | 58.4            | 92.6            | 83.6            | 75.9     |
> | $L_2$-distance | 52.3            | 85.8            | 81.3            | 53.2            | 91.3            | 82.3            | 52.6            | 90.3            | 82.9            | 74.7     |
> | GW             | 42.0            | 71.6            | 70.0            | 41.6            | 71.0            | 69.4            | 42.3            | 71.3            | 70.0            | 61.0     |
> | JS             | **60.0**        | **91.6**        | **83.6**        | **57.4**        | **95.8**        | 87.7            | **59.1**        | **95.2**        | 88.4            | **79.9** |
>
> In Table R4-2, KL-ST and KL-TS denote the KL-divergence $KL(R_k^s, R_l^t)$ and $KL(R_l^t, R_k^s)$ respectively. GW is the Gromov-Wasserstein distance between $ R_k^s$ and $R_l^t$ where the source/target cost is taken as the $L_2$-distance of source/target keypoints. We find that the JS-divergence  achieves the best performance, compared with KL-ST, KL-TS, $L_1$-distance, $L_2$-distance, and Gromov-Wasserstein. We will include this experiment in Appendix B, and cite them in Section 4 in the revised paper.

---

### Official Review · Reviewer_n57r · 2022-06-30

**Rating:** 6
**Confidence:** 3
**Soundness:** 2 fair
**Presentation:** 2 fair
**Contribution:** 2 fair

**Summary:**

This paper utilizes annotated keypoints to tackle incorrect matchings of optimal transport models. First, the authors impose a mask-based constraint on the optimal transport problem to preserve the matching of keypoint pairs. Secondly, they propose to preserve the relation of each data to the keypoints by defining a new cost matrix. Their method, named KeyPoint-Guided model by ReLation preservation (KPG-RL), can be solved by the Sinkhorn’s algorithm and even supported in incomparable spaces. Furthermore, they extend KPG-RL to the partial OT setting. Finally, they verify the effectiveness of their proposal in the heterogeneous domain adaptation application, both close-set and open-set.

**Questions:**

I have the following questions:
1. **Related work**
* The OT problem in “Preservation of matching of keypoints in transport” is similar to the Masked Optimal Transport (Masked OT) [r1]. Please clarify the difference and contribution of this paper compared to the “previous” work.
* In the experimental part, a comparison with Masked OT and Masked GW [r1] can be conducted to illustrate the advantages of the proposed methods. For instance, KPG (w/dist) in Section 5.2 is a good candidate.
[r1] Zhang, Jiying, et al. "Fine-tuning graph neural networks via graph topology induced optimal transport." arXiv preprint arXiv:2203.10453 (2022).

2. **Relation score.** What is the explanation/intuition/inspiration for defining the relation score in Equation 7? Because there are several forms that can satisfy the properties in L170-172. The authors should discuss the choice of the relation score in more detail because, in my opinion, it is one of the major contributions compared to Masked OT.

3. **Cost function.** An ablation study or an explanation for the choice of d, which is set to Jensen-Shannon in this paper, is recommended.

4. **Reproducibility.** A lot of experimental settings are missing and there is no code for reproduction?
* The details of the kernel SVM are missing?
* What is the value of $\alpha$ for each KPG-based method in Section 5.1?
* What is the value of hyper-parameters in Section 5.2?

5. **Experiments.**
* *Sensitivity to $\epsilon$.* The range of $\epsilon$ in Appendix B.2 should be wider, e.g. $\epsilon = 0.001, 0.01, 0.1, 1, \ldots$.
* *Open-set HDA.* Does the set of common labels of the target domain contain or equal the label set of the source domain? If it is the case, it is reasonable to transport more than 1 - $\eta$-proportion (even all) of labeled source domain data. This could be a plausible explanation for the finding in L227-228 in Appendix B.3.
* *Sensitivity to hyper-parameters.* From Tables A-2, A-3, and especially A-5, it is difficult to conclude that KPG-RL is stable to different choices of hyper-parameters.

**Minors**
* In Figure 2, the entry in row 3 and column 6 is 0, which is missing.
* The line numbers in Checklist are outdated.

I am happy to increase my score if the above questions are adequately addressed.



**Limitations:**

The authors stated that the limitation of their method is in L184-185 but I think what they meant was L179-182. Other than that, one additional limitation of KPG-RL is the introduction of new hyper-parameters, which is the temperature $\tau$ and the dissimilarity function $d$. Together with $\epsilon$ in the entropic regularization, users have to tune a lot to find the best set of hyper-parameters.


**Strengths And Weaknesses:**

Though the mask-based constraint version of OT in this paper is not completely new (more details in the next section), the proposed solution, KPG-RL, in this paper is novel and interesting. In terms of the theoretical results, they are simply the extensions of previous works. Moving onto the experimental parts, KPG-RL shows a noticeable improvement over baseline methods in heterogeneous domain adaptation. In addition, the authors also conducted an ablation study to show the stability of their proposal for different choices of hyper-parameters. Overall, this paper is easy to follow with some good illustrations which are really helpful in explaining the concepts. Two major concerns include the lack of references to related works and experimental settings.

---

> ### Author Response · Authors · 2022-08-02
> **Response to Reviewer n57r Part (3/3)**
>
>
> **Q5: On reproducibility and experimental settings/details**
>
> - **Details of kernel SVM.**
>
>   In the kernel SVM, we use the radial basis function kernel $k(x,y)=\exp(-\gamma \|x-y\|^2)$, where $\gamma$ is set to the reciprocal of the feature dimension. We use the scikit-learn packadge of python to implement it by simply running the following codes:
>
>   clf = SVC(gamma='auto')
>    clf.fit(feat_train,label_train)
>
> * **Value of $\alpha$**.
>
>   In this paper, $\alpha$  is simply set to 0.5.
>
> * **The other hyper-parameters in Section 5.2.**
>
>   Apart from $\alpha$, another hyper-parameter is $\epsilon$ , which is set to 0.005.
>
> * **On reproducibility**.
>
>   We will release the source codes for the experiments in this paper on GitHub.
>
> We will include these experimental details in Appendix B.
>
> **Q6: Sensitivity to $\epsilon$.**
>
> The results for varying $\epsilon$ are in the following Table R3-4.
>
> Table R3-4. Results for varying $\epsilon$ .
>
> | $\epsilon$ | 0.0001 | 0.0005 | 0.001 | 0.005 | 0.01 | 0.05 | 0.1  | 1    |
> | ---------- | :------: | :------: | :------: | :------: | :------: | :------: | :------: | :------: |
> | Accuracy   | 83.2   | 83.2   | 83.3  | 83.6  | 82.3 | 76.5 | 74.8 | 71.0 |
>
> **Q7: All labeled source data are transported in Open-set HDA.**
>
> Sorry for making this misunderstanding.  For Open-set HDA, all the labeled source data are transported to target domain. We will clarify this in Section 5.3 in the revised paper.
>
> **Q8: Correcting the statement on the  sensitivity to hyper-parameters.**
>
> Thanks. We will remove the statements that "our method is stable to hyper-parameters."
>
> **Q9: The typos.**
>
> Thanks for the suggestion. We will update Fig. 2 and the Checklist.

---

> > ### Comment · Reviewer_n57r · 2022-08-06
> > **Response to Authors**
> >
> > I thank the authors for their response.
> >
> > After reading their rebuttal, I appreciate that the authors have addressed all of my questions. The authors are encouraged to include the discussion about related work, the experimental details, and additional experiments in the final version of the paper to improve the quality and clarity of the paper.
> >
> > My only remaining concern is that the answer for Q3 is not completely satisfactory. The authors tried to expand what was written in L170-172 and Figure 3, which I understood. From my point of view, Eq. (7) and Eq. (8) strongly resemble the formula of softmax with temperature.
> >
> > *Additional note*: The source code can be zipped and included in the supplementary materials or uploaded anonymously to some websites such as [Anonymous GitHub](https://anonymous.4open.science/).
> >
> > All in all, I would like to increase my score from 4 to 6.

---

> > > ### Author Response · Authors · 2022-08-08
> > > **Response to Reviewer n57r**
> > >
> > > Thanks for the comments again. We will include the discussions on the related work, the experimental details, and the additional experiments in the final version if accepted. Regarding to Q3, built upon the softmax, the relation scores in Eqs. (7) and (8) model the "relation" of each point to the keypoints. Based on the softmax-based formulation, the relation scores in Eqs. (7) and (8) rely on the relative rather than the absolute magnitude of distances, improving their robustness to the scale of distances. This clarification will be included in the paper.
> > >
> > > Our codes can reproduce the results in the paper. The codes contain the extracted features on Office-31 dataset for HDA experiments. We will clean and release the codes with extracted features on the google drive and GitHub.

---

> ### Author Response · Authors · 2022-08-02
> **Response to Reviewer n57r Part (2/3)**
>
>
> **Q3: Explanation of the definition of relation score.**
>
> The relation scores $R_{k,i_u}^s$ and  $R_{l,j_u}^t$ are defined in Eqs. (7) and (8), and illustrated in Fig. 3. Then, based on the relation score, the relation vectors ($R_k^s$ and $R_l^t$) are defined in line 169 of the paper.  We next take the example illustrated in Fig. 3 to explain the definition of the relation score. In Fig. 3, the source point with index $k$ is near to the source keypoint with index $i_2$. The target point with index $l$ is near to the target keypoint with index $j_2$, and $(i_2, j_2)$ are indexes of paired keypoints. According to Eq. (7), $R_{k,i_2}^s$ is close to 1 since $C_{k,i_2}^s$ is much smaller than $C_{k,i_1}^s$ and  $C_{k,i_3}^s$. While $R_{k,i_1}^s$ and $R_{k,i_3}^s$ are close to 0.  As a consequence,  $R_k^s$ is close to $(0,1,0)$. Similarly, $R_l^t$ is also close to $(0,1,0)$. Therefore $G_{k,l}=d(R_k^s,R_l^t)$ could be small.  By the relation preservation model, i.e., KPG-RL model in Eq. (9), the optimal transport plan $M\odot \pi$ has larger entries in the locations where the entries of $G$ are smaller. Hence the cross-domain points corresponding to these locations (e.g. $k$ and $l$ in Fig. 2) that are near to the paired keypoints tend to be matched. Based on the softmax-based formulations in Eqs. (7) and (8),  $d(R_k^s,R_l^t)$ is mainly determined by the relation score to the closest keypoint(s), since relation scores to the distant keypoints are smaller or close to 0. This implies that the points are mainly guided by the closest keypoints in our KPG-RL model in Eq. (9). This explanation will be included in the paragraph under "Modeling the relation to keypoints" in the revised paper.
>
> **Q4: Ablation study and explanation for the choice of $d$.**
>
> Since $R_k^s$ and $R_l^t$ are in the probability simplex, it is reasonable to measure their difference by a distribution divergence/distance. The widely used distribution divergences/distances include the KL-divergence, JS-divergence, and Wasserstein distance. The KL-divergence is not symmetric, so we need to determine the order of inputs. For the Wasserstein distance, one should define the ground metric first. A possible strategy is to set the ground metric to 0 if the two keypoints are paired, otherwise 1.  Such a ground metric makes the Wasserstein distance equal to the $L_1$-distance. In this work, $d$ is taken as the JS-divergence. We compare the performance of different choices of $d$ in the experiment of HDA on Office-31, as in Table R3-3.
>
> Table R3-3. Results of different choices of $d$ in HDA experiment on Office-31.
>
> | Choices of $d$ | A$\rightarrow$A | A$\rightarrow$D | A$\rightarrow$W | D$\rightarrow$A | D$\rightarrow$D | D$\rightarrow$W | W$\rightarrow$A | W$\rightarrow$D | W$\rightarrow$W | Avg      |
> | -------------- | --------------- | --------------- | --------------- | --------------- | --------------- | --------------- | --------------- | --------------- | --------------- | -------- |
> | KL-ST          | 59.0            | 89.7            | **83.6**        | 56.8            | 95.2            | **89.0**        | 57.7            | 93.6            | 88.1            | 79.2     |
> | KL-TS          | 58.1            | 89.0            | 82.3            | 54.2            | 93.9            | 88.1            | 54.2            | 93.2            | **89.4**        | 78.0     |
> | $L_1$-distance | 57.4            | 85.8            | 79.0            | 58.0            | 85.8            | 82.9            | 58.4            | 92.6            | 83.6            | 75.9     |
> | $L_2$-distance | 52.3            | 85.8            | 81.3            | 53.2            | 91.3            | 82.3            | 52.6            | 90.3            | 82.9            | 74.7     |
> | GW             | 42.0            | 71.6            | 70.0            | 41.6            | 71.0            | 69.4            | 42.3            | 71.3            | 70.0            | 61.0     |
> | JS             | **60.0**        | **91.6**        | **83.6**        | **57.4**        | **95.8**        | 87.7            | **59.1**        | **95.2**        | 88.4            | **79.9** |
>
> In Table R3-3, In Table R4-2, KL-ST and KL-TS denote the KL-divergence $KL(R_k^s, R_l^t)$ and $KL(R_l^t, R_k^s)$ respectively. GW is the Gromov-Wasserstein distance between $ R_k^s$ and $R_l^t$ where the source/target cost is taken as the $L_2$-distance of source/target keypoints. We find that the JS-divergence achieves the best performance, compared with KL-ST, KL-TS, $L_1$-distance, $L_2$-distance, and Gromov-Wasserstein. Due to the space limit, we will include this experiment in Appendix B, and cite them in Section 4 in the revised paper.

---

> ### Author Response · Authors · 2022-08-02
> **Response to Reviewer n57r Part (1/3)**
>
>
> We thank the reviewer for the comments. We will revise our paper accordingly.
>
> **Q1: The contribution and difference of the proposed KPG-RL compared with Masked OT [r1].**
>
> Thanks for recommending this related work. Though the paper [r1] of the Masked OT is public as an arXiv paper on March 20, 2022, it is accepted by IJCAI-2022, which is held in July 23-29, 2022, after the NeurIPS submission deadline of May 19, 2022. We did not know this work when working on and submitting this paper. The paper [r1] studies the fine-tuning of the graph neural network (GNN). The authors of [r1] propose the Masked OT model as a regularization term to preserve the local feature invariances between fine-tuned and pretrained GNNs. Our paper investigates the guidance of correct matching in OT using a few annotated keypoint pairs. To impose the guidance of keypoints, we use a mask-based constraint on the transport plan to enforce the matching of keypoint pair in OT. We then propose to preserve the relation (defined in Eqs. (7) and (8)) of each point to the keypoints by our KPG-RL model in Eq. (9). Compared with Masked OT [r1], the research problem of our work is different. In methodology, our main contribution is the relation preservation for imposing the guidance of keypoints, which is different from the work of [r1]. For the mask-based modeling, it is utilized to impose the matching of keypoints, which is theoretically guaranteed by Proposition 1.  While the mask in [r1] aims to preserve the local information of finetuned network from pretrained models. The motivation and the design of the mask in our approach are different from those in Masked OT [r1]. We will cite [r1] in the related work and the mask-based modeling part of the revised paper.
>
> **Q2: Experimental comparison with Masked OT (KP) and Masked GW.**
>
> We first compare the Masked GW using the mask designed in Proposition 1, in HDA experiments on Office-31, shown in Table R3-1.
>
> Table R3-1. Results comparison with Masked GW
>
> |               | A$\rightarrow$A | A$\rightarrow$D | A$\rightarrow$W | D$\rightarrow$A | D$\rightarrow$D | D$\rightarrow$W | W$\rightarrow$A | W$\rightarrow$D | W$\rightarrow$W | Avg      |
> | ------------- | --------------- | --------------- | --------------- | --------------- | --------------- | --------------- | --------------- | --------------- | --------------- | -------- |
> | Masked-GW     | 41.3            | 71.6            | 69.7            | 41.9            | 71.2            | 69.8            | 40.3            | 71.6            | 69.7            | 60.8     |
> | KPG (w/ dist) | 55.2            | 60.7            | 71.6            | 51.3            | 71.9            | 77.1            | 48.7            | 70.0            | 77.7            | 64.9     |
> | KPG-RL        | **60.0**        | **91.6**        | **83.6**        | **57.4**        | **95.8**        | **87.7**        | **59.1**        | **95.2**        | **88.4**        | **79.9** |
>
> From Table R3-1, we can see that KPG-RL outperforms Masked-GW by a margin of 19.2%. Compared with Masked-GW, KPG(w/ dist) only retains the difference between distances from each point to keypoints in the summation of Eq. (3), while Masked-GW contains the difference between distances from each points to all points in the summation of Eq. (3). It is interesting that KPG (w/ dist) performs better than Masked-GW.

---

### Official Review · Reviewer_QaNa · 2022-07-11

**Rating:** 5
**Confidence:** 4
**Soundness:** 3 good
**Presentation:** 4 excellent
**Contribution:** 3 good

**Summary:**

Paper proposes to guide the optimal transport plan by imposing a priori some couplings between "keypoints" into the transport matrix. The relation to each keypoint is also used to define a "guiding" matrix that is considered as the cost matrix of the transport problem. Several variants are proposed, such as a GW- or a partial formulation of the problem. The experimental setup considers the Heteregeneous Domain Adaptation (HDA) scenario, in which it is assumed that some points (of all the classes) of the target distribution are labelled.

**Questions:**

Could you clarify
- why one should use your method than hierarchical OT or TLB?
- could it be extended to the case where the number of keypoints in the source and target distributions are different?

**Strengths And Weaknesses:**

Paper lies in a line of works that aims at constraining the OT problem to encode some extra information or to fasten the computation. Regarding the use of key points, it is quite common to use few anchor points to supervise some of the matching between points or distributions. From a computational point of view, the problem is rewritten by introducing a mask matrix and an algorithm is proposed to solve the problem. To my knowledge, introducing labeled key points to constrain the matching is original. In the context of HDA, experiments show that it is not enough to beat SotA methods. To improve the classification performances, a dedicated cost matrix is constructed. It is based on a function of the distance of each point to each key points within each domain.
The methodology shares similarities with some OT variants, notably with hierarchical OT. In the latter, instead favoring matchings between some keypoints, a matching between subgroups (e.g. clusters) is sought. It also shares some similarities with TLB (« third lower bound » of Gromov-Wasserstein), in which each point of each domain is described as a set of distances to all the points of the same domain, and a « wasserstein of wasserstein » of these distances is computed. The originality and advantages of this « relation preservation » guiding matrix w.r.t. those alternatives is not discussed. No comparison of the performances is neither performed. This constitutes the main weakness of the paper as this guiding matrix seems to be the main ingredient to improve the performances.
The experimental validation of the method is performed on a heterogeneous domain adaptation context. In this context, all the labels from the source distribution are known and only few of the target ones are provided. It is shown that adding only the keypoints to drive the matching is not enough to beat the SotA (10 point behind the best competitor) but that adding the guiding matrix improves the results. Additional experiment shows that performances increase when the number of keypoints. It is unclear if the competitors also use some labeled points to drive the learning, which makes the results with the proposed method and SotA difficult to compare.


Strengths of the paper:
- the paper is really well written and easy to follow; figures illustrate clearly the proposed method.
- the methodology is sound and performs favorably in the considered HDA scenario.
- extensions with entropic regularization or partial OT is provided.

Weaknesses:
- insufficient comparison with similar OT variants such as hierarchical OT and TLB
- experimental setup that does not fit exactly the proposed method: while the hypothesis that we have access to labels of some of the target distribution makes sense, it is unclear how the method is sensitive to the choice of the keypoints within the source domain, or if it could be possible to use all the labeled information of this source domain.
- it is unclear how the setting can be extended to deep method, where mini batches are used.


Minor comments :
- «  Fuzzed OT » should read « fused OT »
- « Frank-Walfe » algorithm
- «  Offce 31 »

---

> ### Author Response · Authors · 2022-08-02
> **Response to Reviewer QaNa Part (3/3)**
>
> **Q4: Could the proposed KPG-RL approach be extended to the case where the number of keypoints in the source and target distributions are different?**
>
> We take the case that the target keypoint number is larger for illustration. In this case, since the keypoints are paired, there may be several target keypoints matched to the same source keypoint. To extend our approach to this situation, we can replace these target keypoints using their centers.  As a result, the keypoints of source and target domains are one-to-one paired. In the experiments in Table 2 of the paper, 2/3 labeled target samples (target keypoints) for one class are given, which should be matched to one source class center (source keypoint). In the implementation, we use the center of these 2/3 labeled target samples as target keypoint, which is matched to the source class center of the same class as the target samples. The results in Table 2 of the paper show that our approach is effective for HDA.
>
> **Q5: Do the competitors use some labeled target samples?**
>
> All the compared HDA methods use the same given labeled target samples. Moreover, all the methods are implemented in five runs. In each run, all the methods use the same training data, including the same labeled source domain data, labeled target domain data, and unlabeled target domain data. We will clarify this in Section 5.2 in the revised paper.
>
> **Q6: The typos.**
>
> Thanks for these suggestions. We will correct them in the revised paper.
>
>
>
> [r1] Nguyen K, Nguyen D, Pham T, et al. Improving mini-batch optimal transport via partial transportation, ICML, 2022.
>
> [r2] Xie S, Zheng Z, Chen L, et al. Learning semantic representations for unsupervised domain adaptation, ICML, 2018.
>
> [r3] Balaji Y, Chellappa R, Feizi S. Robust optimal transport with applications in generative modeling and domain adaptation, NeurIPS, 2020.
>
> [r4] Damodaran B B, Kellenberger B, Flamary R, et al. Deepjdot: Deep joint distribution optimal transport for unsupervised domain adaptation, ECCV, 2018.
>
> [r5] Fatras K, Séjourné T, Flamary R, et al. Unbalanced minibatch optimal transport; applications to domain adaptation, ICML, 2021.

---

> > ### Comment · Reviewer_QaNa · 2022-08-06
> > **Response to authors**
> >
> > First, I would like to thank the authors for their detailed response and the additional figures that have been reported.
> > The impact of the choice of the keypoints (number+location) should be clarified on the final version of the paper, together with a better positionning with the sota (see comments of the other reviewers).
> > That being said, I am happy to raise my score to 6.

---

> > > ### Author Response · Authors · 2022-08-07
> > > **Response to Reviewer QaNa**
> > >
> > > Thanks for the comments again. The impact of the location of the source keypoints has been studied in response to Q2. We next report the results for varying numbers of source keypoints in Table R2-4.
> > > Table R2-4. Results for varying numbers of source keypoints in the HDA experiment on Office-31.
> > >
> > > | Number   | 3    | 5    | 7    | 9    | All  |
> > > | -------- | ---- | ---- | ---- | ---- | ---- |
> > > | Accuracy | 78.4 | 79.2 | 79.6 | 79.8 | 79.9 |
> > >
> > > In this experiment, we randomly sample 3/5/7/9 samples (keypoints) or use all the source samples (keypoints) for each class in the source domain to compute the source class centers, which are paired with labeled target samples for constructing the keypoint pairs. The results in Table R2-4 show that as the number of source keypoints increases, the accuracy gradually increases. The best result is obtained when all source samples are used to compute the class centers.
> > >
> > > As suggested, the impact of the number and location of source keypoints, and the clearer comparison with other SOTA methods will be included in Section 5.2 of the final paper.

---

> ### Author Response · Authors · 2022-08-02
> **Response to Reviewer QaNa Part (2/3)**
>
> Table R2-2. Results for different source keypoints.
>
> | S1   | S2   | S3   | S4   | S5   | Centers  |
> | :----:| :----:| :----:| :----:| :----: | :--------: |
> | 76.8 | 77.5 | 78.2 | 77.8 | 76.9 | **79.9** |
>
> From Table R2-2, we can see that using the class center as the keypoints achieves the best results, compared with randomly sampling one data point per class as the keypoints. This may be because the class centers are estimated using all the data of each class, and these centers can better represent each class than a randomly sampled data point of each class. This experiment will be included in Appendix B and cited in Section 5.2 in the revised paper.
>
> **Q3: Extending the proposed approach to deep learning using mini-batch-based implementation.**
>
> To extend our method to the mini-batch-based implementation, the main challenge is that some of the samples in the mini-batch may not be matched. For instance, in domain adaptation, the categories of some samples in the source mini-batch may not be present in the target mini-batch, and thus these source samples should not be transported/matched. Inspired by [r1] that uses partial OT over the mini-batch data to implement deepJDOT [r2], we use our partial KPG-RL-KP model to partially match the mini-batch data in the training of the deep network.  The partial KPG-RL-KP model is modified from Eq. (13) by replacing $G$ by $\alpha C + (1-\alpha) G$. As an experimental example, we apply the partial KPG-RL-KP to the unsupervised domain adaptation experiment on the Office-Home dataset. We take the source and target class centers of the same class as a keypoint pair. The centers are online updated by exponential moving average in training, same as in [r2]. We use the pseudo labels of target data to update the target class centers, due to the lack of target labels. The protocol is the same as that in [r1].  The batch size is set to 65 and the total transport mass ($s$ in Eq. (13)) is set to 0.6, which are the same as those in [r1]. The results are reported in the following Table R2-3.
>
> Table R2-3. Results for unsupervised domain adaptation.
>
> | Method           | A$\rightarrow$C | A$\rightarrow$P | A$\rightarrow$R | C$\rightarrow$A | C$\rightarrow$P | C$\rightarrow$R | P$\rightarrow$A | P$\rightarrow$C | P$\rightarrow$R | R$\rightarrow$A | R$\rightarrow$C | R$\rightarrow$P | Avg       |
> | ---------------- | --------------- | --------------- | --------------- | --------------- | --------------- | --------------- | --------------- | --------------- | --------------- | --------------- | --------------- | --------------- | --------- |
> | ROT [r3]         | 47.20           | 71.80           | 76.40           | 58.60           | 68.10           | 70.20           | 56.50           | 45.00           | 75.80           | 69.40           | 52.10           | 80.60           | 64.30     |
> | m-OT  [r4]       | 51.75           | 70.01           | 75.79           | 59.60           | 66.46           | 70.07           | 57.60           | 47.88           | 75.29           | 66.82           | 55.71           | 78.11           | 64.59     |
> | m-UOT [r5]       | 54.99           | **74.45**       | **80.78**       | 65.66           | 74.93           | 74.91           | 64.70           | 53.42           | 80.01           | 74.58           | 59.88           | 83.73           | 70.17     |
> | m-POT [r1]       | 55.65           | 73.80           | 80.76           | **66.34**       | 74.88           | **76.16**       | 64.46           | 53.38           | **80.60**       | 74.55           | 59.71           | 83.81           | 70.34     |
> | **m-KPG-RL-KP**  | 52.13           | 63.65           | 74.53           | 61.12           | 67.84           | 67.88           | 59.84           | 52.93           | 76.90           | 71.92           | 59.21           | 82.55           | 65.88     |
> | **m-PKPG-RL-KP** | **57.96**       | **74.45**       | 78.75           | 66.30           | **75.22**       | 74.39           | **66.87**       | **58.47**       | 80.47           | **75.15**       | **61.15**       | **84.23**       | **71.12** |
>
> In Table R2-3, ROT [r3] is a robust OT method. m-OT is the direct mini-batch implementation of deepJDOT [r4]. m-UOT [r5] and m-POT [r1] are respectively unbalanced deepJDOT and partial deepJDOT on mini-batch data. m-KPG-RL-KP is the direct mini-batch implementation of our KPG-RL-KP model. m-PKPG-RL-KP is the mini-batch implementation of our partial KPG-RL-KP model. We can see that by partially matching the samples in the mini-batches,  m-KPG-RL-KP outperforms m-KPG-RL-KP by a margin of 6.24%. Our partial KPG-RL-KP (m-PKPG-RL-KP) outperforms partial DeepJDOT (m-POT) by 0.68%, indicating that using partial matching, our approach is effective for unsupervised domain adaptation under mini-batch implementation. This experiment will be included in Appendix B and cited in Section 5 in the revised paper.

---

> ### Author Response · Authors · 2022-08-02
> **Response to Reviewer QaNa Part (1/3)**
>
> We thank the reviewer for the comments and suggestions. We will revise our paper accordingly.
>
> **Q1: Comparison with hierarchical OT and TLB, and clarify why use the proposed approach than hierarchical OT and TLB.**
>
> The hierarchical OT (HOT) [36] splits the data points into some subgroups/clusters and then derives the matching of these subgroups by OT taking the Wasserstein distances between subgroup pairs as the ground metric. Our approach aims to use the annotated keypoint pairs to guide the matching of other points in OT.  We use a mask-based constraint on the transport plan to enforce the matching of keypoints and impose the guidance of keypoints by preserving the relation to the keypoints. Compared with HOT, the goal of our approach is different. In methodology, we do not explicitly divide the points into subgroups, and there is no hierarchy in our model (see Eq. (9)). The relation score defined in Eqs. (7) and (8) could be treated as the "soft assignment" of each point to keypoints.
>
> TLB [27] is a lower bound of the Gromov-Wasserstein that can be computed faster. TLB takes the ordered distance of each point to all the points in the same domain as features, and then performs the Kantorovich formulation of OT using such features. Differently, our method uses a carefully designed relation (see Eqs. (7) and (8)) of each point to the keypoints to impose the guidance of keypoints to the other points. As shown in experiments, our KPG-RL model outperforms KPG (w/ dist) by a large margin (by 15%) in Table 1, indicating that preserving relation can impose the guidance better than preserving distance.
>
> We next compare our KPG-RL with HOT and TLB in the HDA experiment on Office-31, as in Tabel R2-1.
>
> Tabel R2-1. Results for hierarchical OT, TLB, and KPG-RL in HDA experiment on Office-31.
>
> |            | A$\rightarrow$A | A$\rightarrow$D | A$\rightarrow$W | D$\rightarrow$A | D$\rightarrow$D | D$\rightarrow$W | W$\rightarrow$A | W$\rightarrow$D | W$\rightarrow$W | Avg      |
> | ---------- | --------------- | --------------- | --------------- | --------------- | --------------- | --------------- | --------------- | --------------- | --------------- | -------- |
> | HOT        | 39.0            | 44.8            | 40.0            | 31.3            | 52.6            | 44.8            | 29.7            | 60.0            | 56.5            | 44.3     |
> | TLB        | 29.4            | 36.5            | 43.2            | 24.5            | 31.3            | 51.0            | 23.6            | 31.9            | 49.7            | 35.7     |
> | Masked-HOT | 45.2            | 60.3            | 57.4            | 48.9            | 63.5            | 59.2            | 40.3            | 67.1            | 61.4            | 55.9     |
> | Masked-TLB | 42.5            | 66.3            | 64.7            | 38.5            | 68.5            | 65.9            | 43.1            | 68.2            | 67.3            | 58.3     |
> | **KPG-RL** | **60.0**        | **91.6**        | **83.6**        | **57.4**        | **95.8**        | **87.7**        | **59.1**        | **95.2**        | **88.4**        | **79.9** |
>
> Note that when implementing HOT, we first cluster the target data into 31 clusters using k-means where the centers are initialized by the class centers estimated by labeled target data. Then we perform KP on the clusters, in which the ground metric is the Gromov-Wasserstein distance between each source cluster and each target cluster, since the source and target clusters are in different spaces. In Table R2-1, Masked-HOT and Masked-TLB are respectively the variants of HOT and TLB that use our mask-based modeling of transport plan to enforce the matching of labeled data in HOT and TLB. From Table R2-1, we can observe that, with the mask-based constraint, the performances of HOT and TLB are improved but still largely lower than the performance of our KPG-RL by more than 20%. This confirms the importance of the relation for imposing the guidance of keypoints. The methodology comparison with HOT and TLB will be included in the related work, and the experimental comparison will be included in Section 5.2 in the revised paper.
>
> The above comparison indicates that our method is more suitable than HOT or TLB for the applications that a few paired keypoints could be defined. For these applications, our approach can better use the keypoints to guide the correct matching of the other points.
>
> **Q2: Sensitivity to source keypoints.**
>
> In the experiments of the paper, the source keypoints are taken as the source class centers. To study the sensitivity to the source keypoints, we randomly sample one data point from each class as a keypoint to construct the source keypoints. We run the experiments with five different samplings for constructing the source keypoints (these five runs are denoted as S1, S2, S3, S4, S5 respectively). The results are reported in the following Table R2-2.

---

### Official Review · Reviewer_uAn8 · 2022-07-25

**Rating:** 5
**Confidence:** 3
**Soundness:** 2 fair
**Presentation:** 3 good
**Contribution:** 2 fair

**Summary:**

The paper introduces a semi-supervised OT formulation and its solutions and applied it to solving heterogeneous domain adaptation. The weak supervision comes from limited labeling of a set of key points in source and target domains. The authors constructed a key point-guided model by the doc product of the transport plan masked by the binary key point connection and a relation matrix between source and target. They then appended the model to the OT formulation as a weighted regularization term. Finally, they extended their model to partial OT problems. Experiments on Office-31 showed that the proposed model achieved better results overall than several existing methods.

**Questions:**

---
What's the difference between the idea of key point-guided OT and the idea of semi-surprised OT in OTDA by Courty et al. [9]?

---
Line 138: "If the paired key points (I,j) are matched, ... all mass of point $x_i$ must be transported to $y_j$ and $y_j$ can only receive the mass from $x_i$."

I miss the motivation of this idea. Since we're solving Kantorovich OT, how do we expect the connection between key points and non-key points are one-to-one?

---
Line 161: Why do we set it to $0.1 \times \max_{I,k}\{C_{I,k}^{s}\}$? Is $\tau$ tunable and thus cross-validated?

Where do $R_{k, i_u}^{s}$ and $G_{k,l}=d(R_k^s, R_l^t)$ come from? Why do we choose these discrepancies to construct the relation?

Plus, $G$ would take quite a lot of memory if $m$ and $n$ are large. What is the peak memory consumption during the experiments with Office-31?

**Limitations:**

No negative societal impact found. See above for limitations.

**Strengths And Weaknesses:**

+ The formulations (11-13) are novel to the best of my knowledge.
+ Theorem 1 looks good. The authors solved their model on partial OT problems by adding dummy components to the formulation and later proved that solution to the original problem can be easily derived from the solution to the reformulated problem and the reformulated problem is solvable.

- The new formulation is quite expansive in memory (perhaps in time as well) because of additional pair-wise discrepancies and the authors didn't discuss that at all.
- The authors didn't discuss the difference between their key point-guided formulation and semi-supervised OT. I feel that $L_{kpg}$ can be directly incorporated into $L_{OT}$ by designing a more informative mask $M$ that directly encodes the "relation". It looks redundant to me that the authors had to design a binary mask but then treat it as a regularizer not a constraint. Perhaps it's a dead end but I feel in general the authors didn't clearly explain their motivation and ideas.

---

> ### Author Response · Authors · 2022-08-02
> **Response to Reviewer uAn8 Part (3/3)**
>
> Regarding to $d$, since $R_k^s$ and $R_l^t$ are in the probability simplex, it is reasonable to measure their difference by a distribution divergence/distance. The widely used distribution divergences/distances include the KL-divergence, JS-divergence, and Wasserstein distance. The KL-divergence is not symmetric, so we need to determine the order of inputs. For the Wasserstein distance, one should define the ground metric first. A possible strategy is to set the ground metric to 0 if the two keypoints are paired, otherwise 1.  Such a ground metric makes the Wasserstein distance equal to the $L_1$-distance. In this work, $d$ is taken as the JS-divergence. We compare the performance of different choices of $d$ in the experiment of HDA on Office-31, as in Table R1-4.
>
> Table R1-4. Results of different choices of $d$ in HDA experiment on Office-31.
>
> | Choices of $d$ | A$\rightarrow$A | A$\rightarrow$D | A$\rightarrow$W | D$\rightarrow$A | D$\rightarrow$D | D$\rightarrow$W | W$\rightarrow$A | W$\rightarrow$D | W$\rightarrow$W | Avg      |
> | -------------- | --------------- | --------------- | --------------- | --------------- | --------------- | --------------- | --------------- | --------------- | --------------- | -------- |
> | KL-ST          | 59.0            | 89.7            | **83.6**        | 56.8            | 95.2            | **89.0**        | 57.7            | 93.6            | 88.1            | 79.2     |
> | KL-TS          | 58.1            | 89.0            | 82.3            | 54.2            | 93.9            | 88.1            | 54.2            | 93.2            | **89.4**        | 78.0     |
> | $L_1$-distance | 57.4            | 85.8            | 79.0            | 58.0            | 85.8            | 82.9            | 58.4            | 92.6            | 83.6            | 75.9     |
> | $L_2$-distance | 52.3            | 85.8            | 81.3            | 53.2            | 91.3            | 82.3            | 52.6            | 90.3            | 82.9            | 74.7     |
> | GW             | 42.0            | 71.6            | 70.0            | 41.6            | 71.0            | 69.4            | 42.3            | 71.3            | 70.0            | 61.0     |
> | JS             | __60.0__        | __91.6__        | __83.6__        | __57.4__        | __95.8__        | 87.7            | __59.1__        | __95.2__        | 88.4            | __79.9__ |
>
> In Table R1-4, KL-ST and KL-TS denote the KL-divergence $KL(R_k^s, R_l^t)$ and $KL(R_l^t, R_k^s)$ respectively. GW is the Gromov-Wasserstein distance between $ R_k^s$ and $R_l^t$ where the source/target cost is taken as the $L_2$-distance of source/target keypoints. We find that the JS-divergence achieves the best performance, compared with KL-ST, KL-TS, $L_1$-distance, $L_2$-distance, and Gromov-Wasserstein. Due to space limit, we will include this experiment in Appendix B, and cite it in Section 4.
>
> [r1] Nguyen K, Nguyen D, Pham T, et al. Improving mini-batch optimal transport via partial transportation, ICML, 2022.
>
> [r2] Chen T, Kornblith S, Norouzi M, et al. A simple framework for contrastive learning of visual representations, ICML, 2020.
>
> [r3] Khosla P, Teterwak P, Wang C, et al. Supervised contrastive learning, NeurIPS, 2020.

---

> ### Author Response · Authors · 2022-08-02
> **Response to Reviewer uAn8 Part (2/3)**
>
> **Q3: About encoding the relation in the mask.**
>
> In this paper, we use a mask-based constraint on the transport plan to enforce the matching of keypoints (which is proved in Proposition 1).  As illustrated in Fig. 2, we model the transport plan $\tilde{\pi}$ as the Hadamard product of a mask matrix $M$ and a matrix $\pi$, defined as in Eq. (4). After that, we define the relation score of each point to the keypoints (as in Eqs. (7) and (8)), which is preserved by our KPG-RL model to impose the guidance to the other points. The relation score for each point in Eqs. (7) and (8) is defined based on the distances of the point to all the keypoints, modeling the "relation" of points to the keypoints. We have tried, however, it is non-trivial to model the "relation" to the keypoints by purely designing the mask matrix $M$, and this idea will be left in our future work.
>
> **Q4: Clarifying line 138.**
>
> Thanks for this question. It is true that the Kantorovich problem does not require the one-to-one match. The requirements in line 138 are for the given paired keypoints. However, based on our mask-based OT formulation, the remaining points excluding keypoints are not required to be one-to-one match. Specifically, as we stated in line 133, $\mathcal{K}$ is the set of indexes of the keypoint pairs. For any $(i,j)\in\mathcal{K}$, $i$ and $j$ are respectively the indexes of keypoints $x_i$ and $y_j$ that are paired and should be matched in OT. Therefore,  $x_i$ and $y_j$ satisfy that all mass of point $x_i$ should be transported to $y_j$ and $y_j$  can only receive the mass from $x_i$, which is realized by the mask-based constraint on the transport plan. We will make this clearer in the revised paper.
>
> **Q5: On the choice of $\tau$.**
>
> We set $\tau$ to $\rho*\max_{i,k}C^s_{i,k}$. Therefore, in Eq. (7), the distances are divided by their maximum value, and normalized to $[0,1]$, which may increase the robustness of the relation score to the scale of the distances. $\rho$ is a tunable parameter that controls the "sharpness" of the relation vector $R_k^s$ defined in line 169. If $\rho$ is smaller, all the relation vectors may be closer to "one-hot" vectors.  If $\rho$ is larger, $R_k^s$ may be closer to uniform probability vectors.  In this paper, we empirically set $\rho$ to 0.1,  because 0.1 is a commonly used temperature in the softmax function, e.g., in [r2] and [r3] where the absolute value of the input of softmax is smaller than 1. Please refer to Table A-3 in Appendix B.2 for the effect $\rho$. We will include this explanation in the paragraph under "Modeling the relation to keypoints" in the revised paper.
>
> **Q6: Clarifying the definition of $R_{k,i_u}^s$ and $G_{k,l}=d(R_k^s,R_l^t)$, and the reasons for using these discrepancies to construct the relation preservation model.**
>
> The relation scores $R_{k,i_u}^s$ and  $R_{l,j_u}^t$ are defined in Eqs. (7) and (8), and illustrated in Fig. 3. Based on the relation score, the relation vectors ($R_k^s$ and $R_l^t$) are defined in line 169 of the paper. We next take the example illustrated in Fig. 3 to explain the definition of the relation score. In Fig. 3, the source point with index $k$ is near to the source keypoint with index $i_2$. The target point with index $l$ is near to the target keypoint with index $j_2$, and $(i_2, j_2)$ are indexes of paired keypoints. According to Eq. (7), $R_{k,i_2}^s$ is close to 1 since $C_{k,i_2}^s$ is much smaller than $C_{k,i_1}^s$ and  $C_{k,i_3}^s$. While $R_{k,i_1}^s$ and $R_{k,i_3}^s$ are close to 0.  As a consequence, $R_k^s$ is close to $(0,1,0)$. And similarly, $R_l^t$ is close to $(0,1,0)$. This implies that $R_k^s$ and $R_l^t$ could be similar, and then $G_{k,l}=d(R_k^s,R_l^t)$ could be small.  By the relation preservation model, i.e., KPG-RL model in Eq. (9), the optimal transport plan $M\odot \pi$ has larger entries in the locations where the entries of $G$ are smaller. Hence the cross-domain points corresponding to these locations (e.g. $k$ and $l$ in Fig. 2) that are near to the paired keypoints tend to be matched. Based on the softmax-based formulations in Eqs. (7) and (8),  $d(R_k^s,R_l^t)$ is mainly determined by the relation score to the closest keypoint(s), since relation scores to the distant keypoints are small or close to 0. This implies that the points are mainly guided by the closest keypoints in our KPG-RL model in Eq. (9). We will include this explanation in the paragraphs under "Modeling the relation to keypoints" and "Keypoint-guided model" in the revised paper. (The rest response is in part (3/3))

---

> ### Author Response · Authors · 2022-08-02
> **Response to Reviewer uAn8 Part (1/3)**
>
> We thank the reviewer for the comments. We will revise our paper accordingly.
>
> **Q1:  Memory and time cost of KPG-RL, and the peak memory consumption in the experiments with Office-31.**
>
> We report the memory and time cost of KPG-RL with different sizes of the guiding matrix $G$ in the bottom row of Tables R1-1 and R1-2 respectively. For comparisons, we also report the memory and time cost of the Kantorovich Problem (KP). KP needs to calculate the pair-wised cost matrix $C$, as in Eq. (1). KPG-RL calculates the relation score defined in Eqs. (7) and (8), and then computes the guiding matrix $G$, as in line 174.  Since we have deduced Sinkhorn's algorithm for solving KPG-RL, we solve both KP and KPG-RL using Sinkhorn's algorithm with $\epsilon=0.005$. Table R1-1 shows that KPG-RL costs a slightly larger memory than KP. Table R1-2 shows the computational time for solving KPG-RL and KP problems.
>
> Table R1-1. __Peak memory__ for computing KP and KPG-RL.
>
> | Size ($m\times n$) of $C$/$G$ | $500\times 500$ | $1000\times 1000$ | $2000\times 2000$ |
> | ----------------------------- | :-------------: | :---------------: | :---------------: |
> | **KP**                        |      201M       |       218M        |       330M        |
> | **KPG-RL**                    |      207M       |       232M        |       378M        |
>
> Table R1-2. __Time cost__ for computing KP and KPG-RL.
>
> | Size ($m\times n$) of $C$/$G$ | $500\times 500$ | $1000\times 1000$ | $2000\times 2000$ |
> | ----------------------------- | :-------------: | :---------------: | :---------------: |
> | **KP**                        |      6.9s       |       27.7s       |       60.1s       |
> | **KPG-RL**                    |      10.8s      |       42.1s       |       76.5s       |
>
> In the experiment on Office-31, the maximum memory of $G$ is 38M, and the peak memory of the running process is 780M.  The discussion on memory and time cost will be included in Appendix B and cited in Section 5.2 of the revised paper.
>
> **Q2: Difference of ideas of the proposed keypoint-guided OT to the semi-supervised OT proposed in [9].**
>
> The main difference of our keypoint-guided OT to the semi-supervised OT proposed in [9] is on the formulation of guidance of keypoints to the other points. Our keypoint-guided OT aims to use a few annotated keypoint pairs to guide the correct matching of the other points in OT. To realize this goal, we first use a mask-based constraint on the transport plan to enforce the matching of keypoints pairs, as illustrated in Fig. 2. We then preserve the relation (defined in Eqs. (7) and (8)) of each point to the keypoints to impose the guidance of keypoints. The semi-supervised OT [9] constrains the cost function to encourage the matching of labeled data across source and target domains that share the same class labels. [9] uses the Laplacian regularization to preserve the data structure, different from our keypoint-guided OT that explicitly models the guidance of keypoints matching to the other data points in our OT formulation, optimized by Sinkhorn's algorithm. We experimentally compare the matching accuracy using the toy data in Section 5.1 of the paper. The matching accuracy is computed as in lines 191-194 in the Appendix.
>
> Table R1-3. Matching accuracies of semi-supervised OT [9] and KPG-RL-KP on toy data.
>
> | Number of keyponit pairs | 0        | 2        | 3        | 20      | 30      |
> | ------------------------ | -------- | -------- | -------- | ------- | ------- |
> | Semi-supervised OT [9]   | **41.7** | 41.7     | 41.7     | 58.3    | 66.7    |
> | KPG-RL-KP                | **41.7** | **81.7** | **96.7** | **100** | **100** |
>
> From Table R1-3, it can be observed that given a few (2 or 3) labeled keypoint pairs, our proposed KPG-RL-KP can apparently improve the matching accuracy (by 40% or 55%). While the semi-supervised OT [9] can not improve the matching performance with 2 or 3 labeled keypoint pairs. When the number of labeled keypoint pairs increases (to 20 or 30), the matching performance of semi-supervised OT [9] is improved, but worse than our approach (by more than 30%).
> We will include the methodology comparison in the related work of the revised paper, and the experimental comparison in Appendix B.

---

### Meta-Review · Area_Chair_aV93 · 2022-08-24

**Recommendation:** Accept
**Confidence:** Less certain

**Metareview:**

In this paper the authors propose a novel Optimal Transport problem that uses a small number of annotated  keypoints in both source and target domain to encode additional information and guide the OT plan in the problem. The authors propose a variant of the sinkhorn algorithm to solve the problem and show that it can be used to solve OT across different spaces with also an extension to Partial OT setting. Numerical experiments show the interest of the method on a difficult heterogeneous domain adaptation problem.

The contribution was appreciated and all reviewers agree about the novelty of the method and the interest of the new model in practical applications. All experiments (in the paper, appendix and the new ones in reply) show that the method work better than existing approaches in semi-supervised HDA. Some concerns were raised by reviewers: missing discussion and citation of Masked OT and other approaches such as Fused GW but those were mostly addressed in the replies. The question of the choice of d was also well answered with new experiments.

The consensus between reviewers was that the replies were great and that the paper should be accepted an NeurIPS.  Nevertheless it was very clear from the discussion that the new results, discussion and positioning wrt the state of the art MUST be included in the final version and its supplementary.  The paper is also lacking a discussion about how to obtain keypoints pairs in practice (other than using existing labels) which is very important to ensure that the method can be used in practice.


**Award:**

No

---

### Decision · Program_Chairs · 2022-09-14

Accept